# ROBUST ADAPTIVE MULTI-STEP PREDICTIVE SHIELD-ING

**Tanmay Ambadkar**[*]
The Pennsylvania State University
University Park, PA 16802, USA
tsa5252@psu.edu

**Darshan Chudiwal**
The Pennsylvania State University
University Park, PA 16802, USA
dsc5636@psu.edu

**Greg Anderson**
Reed College
Portland, OR 97202, USA
grega@reed.edu

**Abhinav Verma**
The Pennsylvania State University
University Park, PA 16802, USA
verma@psu.edu

## ABSTRACT

Reinforcement learning for safety-critical tasks requires policies that are both high-performing and safe throughout the learning process. While model-predictive shielding is a promising approach, existing methods are often computationally intractable for the high-dimensional, nonlinear systems where deep RL excels, as they typically rely on a patchwork of local models. We introduce RAMPS, a scalable shielding framework that overcomes this limitation by leveraging a learned, linear representation of the environment's dynamics. This model can range from a linear regression in the original state space to a more complex operator learned in a high-dimensional feature space. The key is that this linear structure enables a robust, look-ahead safety technique based on a *multi-step Control Barrier Function (CBF)*. By moving beyond myopic one-step formulations, RAMPS accounts for model error and control delays to provide reliable, real-time interventions. The resulting framework is minimally invasive, computationally efficient, and built upon robust control-theoretic foundations. Our experiments demonstrate that RAMPS significantly reduces safety violations compared to existing safe RL methods while maintaining high task performance in complex control environments.

## 1 INTRODUCTION

Deep reinforcement learning (RL) has achieved remarkable success in solving complex control problems, yet its deployment in safety-critical applications like autonomous vehicles and robotics remains a grand challenge Gu et al. (2022). A core requirement in these domains is not only that the final policy is safe, but that safety is maintained throughout the entire learning process. This problem of *safe exploration* has motivated a range of solutions, among which model-predictive shielding has emerged as a promising paradigm Jovanovi'c et al. (2020); Brunke et al. (2021).

Existing shielding frameworks present a difficult trade-off. On one hand, neural shields learn safety critics from data, offering flexibility but often requiring vast experience and failing to prevent violations during early training Bharadhwaj et al. (2021b); Dalal et al. (2018). On the other hand, symbolic shields provide formal, mathematical guarantees from the first interaction by analyzing an environment model Berkenkamp et al. (2017); Anderson et al. (2020); Wang & Zhu (2024). However, these methods have a critical limitation that has confined them to low-dimensional systems: they rely on explicitly partitioning the state space to construct a patchwork of local linear models. This approach suffers from the curse of dimensionality, rendering it computationally intractable for the complex, high-dimensional environments ($> 10$ dimensions) where modern deep RL excels.

This paper introduces RAMPS, a framework that bridges this critical gap by making formal shielding scalable to high-dimensional, nonlinear systems through a novel theoretical advance in safety cer-

---

[*]Website: https://ambadkar.com/ramps

tification. At the core of RAMPS is a new robust multi-step Control Barrier Function formulation that fundamentally changes how safety is guaranteed in discrete-time stochastic systems with model uncertainty. RAMPS achieves both theoretical soundness and practical scalability through a unified approach.

The key insight enabling RAMPS is the synergy between our robust multi-step CBF theory and the use of linear dynamics models. By representing the system dynamics through a single linear model, whether a linear regression in the original space or a learned operator in a high-dimensional feature space like the Deep Koopman OperatorShi & Meng (2022), we can efficiently propagate safety constraints multiple steps into the future while formally accounting for model error. Our CBF formulation explicitly incorporates accumulated prediction error through a novel tightening mechanism, provides model-relative safety guarantees even with imperfect models. At each timestep, RAMPS's shield solves a comparatively-small Quadratic Program to find the minimally invasive safe action, with adaptive horizon selection that maximizes foresight while avoiding excessive conservatism.

Our contributions are threefold:

- We introduce RAMPS, a scalable shielding framework that provides probabilisitic safety guarantees, in high-dimensional, nonlinear systems by unifying robust CBF theory with learned linear dynamics representations.

- We develop a novel robust multi-step CBF formulation for discrete-time stochastic systems featuring accumulated error tightening and adaptive horizon selection, providing a principled solution to high relative-degree safety constraints under model uncertainty.

- We demonstrate that RAMPS significantly outperforms state-of-the-art safe RL methods, reducing safety violations by up to 90% and scaling to 348-dimensional environments, while maintaining competitive task performance across challenging high-dimensional control environments including quadrupedal locomotion.

## 2 RELATED WORK

Research in safe reinforcement learning (safe RL) can be categorized by *what kind of safety guarantees are provided* and *when those guarantees apply*. Safety is usually defined in two ways: (i) a *cost-based formulation*, where each action may incur some penalty and the long-term cost must remain below a threshold, or (ii) a *state-based formulation*, where specific regions of the state space are marked unsafe and must never be entered. Our work adopts the state-based view.

**Worst-Case Guarantees.** One line of work provides *deterministic safety guarantees* under a worst-case environment model, ensuring forward invariance by construction (Anderson et al., 2020; Gillula & Tomlin, 2012; Alshiekh et al., 2018; Zhu et al., 2019; Fulton & Platzer, 2019; Bacci et al., 2021). These approaches offer strong guarantees but require an explicit model and are limited to low-dimensional settings due to the computational cost of state-space partitioning. In contrast, RAMPS does not require a predefined model and remains tractable in high-dimensional systems.

**Statistical Guarantees.** Another family of methods offers *probabilistic or statistical safety guarantees*. These approaches build or learn an approximate dynamics model and optimize policies that are *likely* to be safe with respect to that model (Achiam et al., 2017; Liu et al., 2020; Yang et al., 2020; Ma et al., 2021; Zhang et al., 2020; Satija et al., 2020). While more scalable than worst-case methods, they typically allow safety violations during training. In contrast, RAMPS enforces hard constraints with respect to its learned model, reducing violations in practice.

**Model-Predictive Shielding.** A complementary paradigm is *model-predictive shielding* (MPS), where a shield monitors the agent's proposed action and intervenes only when it threatens safety (Wabersich & Zeilinger, 2018; Bastani, 2021; Anderson et al., 2020; 2023; Goodall & Belardinelli, 2023; Banerjee et al., 2024). Prior works differ in how they construct models and shields, but most struggle with scalability, particularly when moving beyond simple one-step predictions. Model-predictive shielding is closely related to **model predictive control (MPC)**: both use a model to roll out multi-step trajectories and solve constrained optimization problems. However, their objectives differ fundamentally. MPC optimizes long-horizon performance and effectively replaces the policy with its own control solution, whereas MPS acts purely as a *safety filter*: it retains the agent's action

whenever it is safe, and otherwise solves a feasibility problem to return the closest safe alternative. This shifts the role of prediction from planning to minimal, targeted intervention, making shielding compatible with arbitrary RL policies while still enforcing hard safety guarantees.

**Koopman Operators and Safety.** Prior work has combined Koopman models with safety mechanisms, typically through *one-step* CBF filters. This includes Koopman-accelerated backup-CBF controllers, and neural or deep approaches that learn Koopman embeddings together with one-step CBF-QP filters or command governors Folkestad et al. (2020); Zinage & Bakolas (2022); Chen et al. (2024); Mitjans et al. (2024); Liang et al. (2025). Robust Koopman-MPC methods provide predictive control with error guarantees Mamakoukas et al. (2022); de Jong et al. (2024). However, these methods either assume a known backup controller, rely on SMT-based CBF certification, or remain limited to one-step filtering and moderate-dimensional systems.

**Cost-Based Safe RL.** Cost-based methods enforce safety indirectly by shaping the reward with carefully designed cost signals and applying constrained optimization techniques (Achiam et al., 2017; Sootla et al., 2022a; Gu et al., 2024; Sootla et al., 2022b; Zhang et al., 2022; Yang et al., 2022). These approaches are flexible but inherently allow violations while the agent learns the cost structure. Compared to these methods, RAMPS enforces stricter state-based safety constraints, leading to fewer violations.

Safe RL methods balance a trade-off: *formal and symbolic methods* offer strong guarantees but do not scale, while *statistical and cost-based methods* scale but permit many violations early in training. Existing model-predictive shielding methods are typically limited to systems with state dimensions in the tens, as they rely on computationally expensive state-space partitioning or explicit nonlinear model propagation. RAMPS bridges this gap by combining a learned, linear model with a novel robust multi-step control barrier function, enabling scalable shielding with strong safety assurances in complex, high-dimensional environments, successfully operating on systems with over 300 state dimensions (where current formal techniques struggle above 10-dimensions), while maintaining real-time computational efficiency.

## 3 PRELIMINARIES

**Safe Exploration.** We model the environment as a Markov decision process (MDP) $\mathcal{M} = (\mathcal{S}, \mathcal{A}, r, P, \gamma)$, where $\mathcal{S}$ is the state space, $\mathcal{A}$ is the action space, $r : \mathcal{S} \times \mathcal{A} \to \mathbb{R}$ is a reward function, $P(x' \mid x, a)$ is a probabilistic transition function, and $p_0$ is an initial distribution over states. A *policy* $\pi$ maps states to distributions over actions. The long-term return of a policy is $R(\pi) = \mathbb{E}_{s_i, a_i \sim \pi}\left[\sum_{i=0}^{\infty} \gamma^i r(s_i, a_i)\right]$. The goal of RL is to find an optimal policy $\pi^* = \arg\max_\pi R(\pi)$.

Most deep RL algorithms generate a sequence of policies $\pi_0, \pi_1, \ldots, \pi_N$ with $\pi_N \to \pi^*$. We refer to this sequence as a *learning process*. In *safe exploration*, the aim is to ensure that every intermediate policy remains safe with high probability. Formally, given a safety threshold $\delta$ and unsafe set $\mathcal{S}_U$, we require $\forall\, 1 \leq i \leq N, \ \Pr_{s \sim \pi_i}(s \in \mathcal{S}_U) \leq \delta$, while the final policy $\pi_N$ maximizes reward among all safe policies. Following prior work (Anderson et al., 2023; Wang & Zhu, 2024), we do not require the initial policy $\pi_0$ to be safe, since no prior model of the environment is assumed.

**Safety Specification.** We adopt the common state-based notion of safety in safe RL. The unsafe set $\mathcal{S}_U$ is defined as a union of convex polyhedra over features of the state space (Anderson et al., 2023; Wang & Zhu, 2024). Equivalently, the safe set can be expressed as $\mathcal{S} \setminus \mathcal{S}_U = \bigcup_{i=1}^{M} \{\, s \in \mathcal{S} \mid G_i s \leq h_i \,\}$, for matrices $G_i$ and vectors $h_i$. Unions of convex polyhedra are sufficient to approximate any compact safe set to arbitrary precision, and are widely used in model-predictive safety methods.

## 4 ROBUST ADAPTIVE MULTI-STEP PREDICTIVE SHIELDING

RAMPS, provides strong, real-time safety guarantees for reinforcement learning agents by integrating a learned, linear dynamics model with a robust, certificate-based safety shield. The framework is composed of three core components: (1) a learned linear dynamics model that provides a single, global representation of the environment's dynamics from data; (2) a Robust Control Barrier Function

(CBF) that uses this model to certify safety and correct potentially unsafe actions online; and (3) a standard deep RL agent that learns a high-performance policy inside the protection of the shield.

The key requirement for the dynamics model is that it must be linear, as this structure enables the efficient, multi-step evolution required by RAMPS. This allows for a flexible range of modeling choices, from simple linear regression operating in the original state space to the Koopman Operator (Shi & Meng, 2022) that learns a linear transition function in a high-dimensional feature space.

RAMPS operates in an iterative loop. The agent first collects a dataset of environment interactions. This data is used to train the linear dynamics model and a worst-case error bound, which in turn parameterize the CBF shield. The RL agent is then trained, with every action being verified and potentially corrected by the shield to ensure safety. The newly collected, safe data is added back to the dataset, allowing the dynamics model and error bound to be periodically refined. This creates a cycle where a more accurate model leads to a less conservative shield, allowing the agent to explore more freely and learn a better policy. This is illustrated in Algorithm 1.

## 4.1 SAFETY SHIELDING WITH MULTI-STEP ROBUST CONTROL BARRIER FUNCTIONS

We propose a safety shield designed to address a fundamental limitation of standard Control Barrier Functions (CBFs) when applied to discrete-time stochastic systems. Although one-step CBFs offer strong guarantees in continuous time, their discrete-time analogues may fail when a system's control inputs do not immediately affect the safety constraints; a challenge characterized by a relative degree greater than one. To resolve this issue, we construct a *multi-step robust CBF* by drawing upon principles from the theories of High-Order CBFs (HOCBFs; Tan et al. (2022)) and multi-step predictive control (Chriat & Sun, 2023). Our shield enforces safety over a variable prediction horizon $H$, which ensures that control authority is maintained despite such actuation delays. By adaptively selecting the largest feasible horizon at each timestep, the shield maximizes its predictive capability to eliminate "trap" states, which are configurations that appear safe in the short term but lead to inevitable future violations. This is accomplished while remaining minimally invasive to the actions proposed by the reinforcement learning agent's policy.

**Control Barrier Functions (CBFs)** (Nagumo, 1942; Prajna & Jadbabaie, 2004; Wieland & Allgöwer, 2007; Ames et al., 2019) are a powerful tool for enforcing safety constraints in control systems by rendering a specific region of the state space forward invariant. In the **continuous-time** setting, for a system with dynamics $\dot{x} = f(x) + g(x)u$ and a safe set defined as $\mathcal{C} = \{x \in \mathbb{R}^n \mid h(x) \geq 0\}$, a function $h$ is a CBF if there exists a class-$\mathcal{K}$ function $\alpha$ such that for all $x \in \mathcal{C}$, the condition $\sup_{u \in U} [L_f h(x) + L_g h(x)u + \alpha(h(x))] \geq 0$ holds. This Lie derivative condition ensures that for any state on the boundary of the safe set, there exists a control action that prevents the system from instantaneously exiting $\mathcal{C}$.

In contrast, for a **discrete-time** system $x_{k+1} = F(x_k, u_k)$, the condition is fundamentally different. A function $h$ is a discrete CBF if for all $x_k \in \mathcal{C}$, there exists a control $u_k \in U$ such that $h(F(x_k, u_k)) \geq \lambda h(x_k)$, where $\lambda \in [0, 1]$ is a decay rate. The key distinction lies in their temporal nature: the continuous condition is infinitesimal, guaranteeing safety based on the instantaneous velocity of the system, while the discrete condition provides a guarantee over a finite time step, ensuring that the state at step $k + 1$ remains safe given the state at step $k$. This often makes the discrete condition more conservative, as it must account for the system's evolution over the entire sampling period. Reinforcement learning typically deals with discrete-time systems.

**Linear Dynamics.** The core of our shielding framework relies on a learned, linear dynamics model, as this structure is essential for performing the efficient, multi-step predictions needed for robust safety analysis. For systems with simple dynamics, this can be a direct linear model operating in the original state space. For more complex, non-linear environments, the state can be "lifted" via a learned, non-linear embedding into a higher-dimensional feature space (Shi & Meng, 2022). The fundamental principle is that within this lifted space, the intricate dynamics can be accurately captured by a simple linear transition, $z_{k+1} = Az_k + Bu_k + c$. This transformation from non-linear to linear dynamics is what enables the shield to efficiently propagate safety constraints far into the future, making the approach scalable to a wide range of complex systems.

**Safe Set and Dynamics.** Let the lifted state space be $\mathbb{R}^{n+d}$ ($d \geq 0$) with discrete-time affine dynamics $z_{k+1} = Az_k + Bu_k + c + w_k$, where $c$ is a learned constant offset representing the

system's drift, and $w_k$ is an additive model error satisfying $\|w_k\|_\infty \leq \varepsilon$. The admissible control set is $\mathcal{U} \subset \mathbb{R}^m$. We define a polyhedral safe set $\mathcal{C}$ as the intersection of half-spaces, such that

$$\mathcal{C} = \bigcap_{i=1}^M \{ z \mid p_i^\top z + b_i \leq 0 \}.$$

For each face $i$ of the polyhedron, we define a corresponding safety function $h_i(z)$ as $h_i(z) = -(p_i^\top z + b_i)$, which means the safe set can be expressed as $\mathcal{C} = \{ z \mid h_i(z) \geq 0, \ \forall i \}$.

**One-Step Robust CBF Condition.** To guarantee safety under model uncertainty, we formulate a robust CBF condition similar to (Cosner et al., 2023). The safety requirement is that the true next state, $z_{k+1} = Az_k + Bu_k + c + w_k$, must remain in the safe set $\mathcal{C}$. This implies that for each face $i$, the condition $p_i^\top(Az_k + Bu_k + c + w_k) + b_i \leq 0$ must hold for bounded disturbances $w_k$.

To ensure this, we design the constraint based on the worst-case disturbance, which has a value of $\varepsilon\|p_i\|_1$. By incorporating this worst-case term, we arrive at the robust CBF condition: for any state $z \in \mathcal{C}$, there must exist a control input $u \in \mathcal{U}$ such that

$$p_i^\top(Az + Bu + c) + b_i \ \leq \ \lambda(p_i^\top z + b_i) \ - \ \varepsilon\|p_i\|_1, \quad \forall i, \tag{1}$$

where $\lambda \in (0, 1]$ is a decay parameter that governs the conservatism of the barrier condition. Values of $\lambda$ close to 1 require the safety function $h_i(z)$ to remain nearly constant across timesteps, leading to stricter constraints and stronger invariance. Smaller values of $\lambda$ relax this requirement by permitting $h_i(z)$ to decay over time, which can improve feasibility but reduces the safety margin. The term $-\varepsilon\|p_i\|_1$ provides an additional robust margin, ensuring safety under the worst-case model error.

**Relative Degree.** The relative degree of a safety constraint $h(z)$ under dynamics $z_{k+1} = f(z_k) + g(z_k)u_k$ is the smallest integer $r \geq 1$ such that the control input $u_k$ appears explicitly in the $r$-step evolution of $h(z_k)$, i.e. through $\frac{\partial h(z_{k+r})}{\partial u_k} \neq 0$.

**Multi-Step Robust CBF Condition.** The one-step condition in equation 1 is insufficient for systems where the control input has a delayed effect on a safety constraint (i.e., relative degree $r > 1$) 4.2. To eliminate the *trap* states that arise in such systems, our method ensures that the safety condition is met at *every* intermediate timestep $j$ over a chosen horizon $H$, for all $j \geq r_i$. For each such step $j$, we define the nominal reachable state under a control sequence $\mathbf{u} = (u_0, \ldots, u_{H-1})$ as

$$z_j(z, \mathbf{u}) \ = \ A^j z + \sum_{k=0}^{j-1} A^{j-1-k} Bu_k + \sum_{k=0}^{j-1} A^k c,$$

where the final term represents the cumulative effect of the affine drift. The total accumulated error over this $j$-step horizon is bounded by a tightening term, $\mathcal{E}_j(p_i)$, which sums the worst-case error at each step:

$$\mathcal{E}_j(p_i) \ = \ \sum_{k=0}^{j-1} \varepsilon \|p_i^\top A^k\|_1.$$

This leads to a set of robust CBF conditions, one for each valid step $j$ and face $i$:

$$p_i^\top z_j(z, \mathbf{u}) + b_i \ \leq \ \lambda^j(p_i^\top z + b_i) \ - \ \mathcal{E}_j(p_i). \tag{2}$$

Each of these inequalities is linear with respect to the full control sequence $\mathbf{u}$. We aggregate all such constraints into a single system of linear inequalities, $G\mathbf{u} \leq h$, which guarantees that any feasible control sequence maintains the system within the safe set $\mathcal{C}$ throughout the entire horizon.

**Minimally Invasive Action Selection.** For a horizon $H$, the shield solves a Quadratic Program (QP) to find a safe control sequence that is minimally invasive to the RL agent's intended action, $a_\pi$. The primary objective is to find a control sequence $\mathbf{u} = (u_0, \ldots, u_{H-1})$ that minimizes the deviation of the first action, $u_0$, from the agent's proposal:

$$\min_{\mathbf{u}} \quad \|u_0 - a_\pi\|_2^2 \tag{3}$$
$$\text{s.t.} \quad G\mathbf{u} \leq h, \quad \text{(representing all constraints from equation 2),}$$
$$\quad u_k \in \mathcal{U}, \quad k = 0, \ldots, H-1.$$

Following the receding horizon principle, only the first action of the solution, $u_0$, is applied to the system. The subsequent actions, $u_{1:H-1}$, are optimized to ensure a feasible trajectory exists but are discarded, preserving flexibility at the next timestep.

**Adaptive Horizon Selection and Safety Guarantee.** At each timestep, we select the horizon $H$ via a bounded binary search within $[H_{\min}, H_{\max}]$, where $H_{\min}$ is the maximum relative degree among active constraints. Candidate horizons are tested by solving the QP in equation 3: feasible horizons remain candidates while the search continues toward larger values, and infeasible ones shrink the range. The largest feasible horizon $H^*$ determines the minimally invasive action $u_0$.

If no feasible horizon is found, a backup policy $u_{\text{backup}}(z)$ (A.4) is applied. Otherwise, the chosen action $u_0$ guarantees forward invariance: under disturbances $\|w_k\|_\infty \leq \varepsilon$, the closed-loop system satisfies $z_k \in \mathcal{C}, \quad \forall k \leq H$.

## 4.2 ANALYSIS OF THE SHIELDING FRAMEWORK

The efficacy of our framework stems from the powerful synergy between a learned linear dynamics model and the multi-step robust CBF shield. Each component is designed to address a fundamental challenge in safe control, and their integration yields a solution that is formally sound, robust to model error, and computationally tractable.

**Synergy of a Linear Model and Multi-Step Shielding.** The foundational element of our approach is the use of a linear dynamics model. This structure is the key enabler for our multi-step shield; it allows safety constraints, defined as simple polyhedra, to be accurately and efficiently propagated through time. Unlike methods that rely on repeated local linearizations or computationally expensive nonlinear propagation, our approach maintains tractability even over extended prediction horizons. This synergy is critical: the linear model makes multi-step prediction feasible, and the multi-step prediction is what gives the shield its foresight and power.

**Robustness to Model Error.** A core design principle of our framework is that it does not assume a perfect dynamics model. Instead, it achieves robustness by formally accounting for model error. The shield's safety guarantee is not based on the model's nominal prediction alone, but on a worst-case analysis that considers the maximum possible deviation. The robust tightening term, $\mathcal{E}_j(p_i)$, is derived from a data-driven error bound $\varepsilon$, effectively creating a *tube* of uncertainty around the predicted trajectory. By ensuring this entire tube remains within the safe set, the shield remains effective even when the learned linear model is an imperfect approximation of the true, complex dynamics. This allows the framework to work well even with simple models like linear regression, as it plans for their inherent inaccuracies.

**Illustrative Example: Resolving High Relative-Degree Traps in Pendulum.** The multi-step CBF framework also addresses traps in systems where the safety constraint depends on a state that the control input does not influence in a single step. Consider the pendulum environment with state $z = (\theta, \omega)$, representing angle and angular velocity. Its dynamics can be written in affine form as

$$z_{k+1} = Az_k + Bu_k + c(z_k), \text{ with } A = \begin{bmatrix} 1 & \Delta t \\ 0 & 1 \end{bmatrix}, \quad B = \begin{bmatrix} 0 \\ \frac{3\Delta t}{m\ell^2} \end{bmatrix}, \quad c(z_k) = \begin{bmatrix} 0 \\ \frac{g\Delta t}{2\ell} \sin(\theta_k) \end{bmatrix}.$$

Suppose we impose a safety constraint on the angle, $p^\top z + b = \theta + \delta \leq 0$, with normal $p = [1\ 0]^\top$. The influence of the control input in one step is determined by $p^\top B$, which evaluates to 0. Thus, a one-step CBF cannot act directly on $\theta$ to prevent it from exceeding the bound. This creates a *relative-degree trap*: the shield has no immediate authority over the constrained variable.

In contrast, our multi-step formulation evaluates terms such as $p^\top A^{k-1}B$. For the pendulum, $p^\top AB = [1\ 0] \begin{bmatrix} \frac{3\Delta t^2}{m\ell^2} \\ \frac{3\Delta t}{m\ell^2} \end{bmatrix} = \frac{3\Delta t^2}{m\ell^2} \neq 0$. This non-zero term indicates that the control input does affect $\theta$, but only after two steps. By enforcing constraints over a horizon $H \geq r$ (here, $r = 2$), our framework ensures that the control authority is accounted for, thereby resolving the trap. The affine term $c(z_k)$ shifts the dynamics but does not alter the relative-degree analysis. This mirrors the role of High-Order Control Barrier Functions in continuous-time systems (Tan et al., 2022; Chriat & Sun, 2023).

**Conditional Safety Guarantees** The safety guarantee provided by our framework is a probabilistic certificate, which is standard for systems with learned dynamics. The argument is twofold: we first

establish a deterministic guarantee of safety relative to our learned model and its error bound, and then connect this guarantee to the true physical system with a probabilistic bound.

**Guarantee Relative to the Learned Model.** Let the true, unknown, discrete-time dynamics of the system be governed by the function $F : \mathcal{S} \times \mathcal{A} \rightarrow \mathcal{S}$, such that the true next state is $s_{k+1} = F(s_k, u_k)$. Our framework learns a linear model, which we denote as $\hat{F}$, that approximates these dynamics in a lifted space.

$$\hat{z}_{k+1} = \hat{F}(z_k, u_k) = A z_k + B u_k + c.$$

The residual dynamics, or one-step prediction error, is the difference between the true evolution of the lifted state and the model's prediction, denoted by $w_k = z_{k+1} - \hat{z}_{k+1}$. Our shield is constructed using the model $\hat{F}$ and a worst-case bound on this error, $\|w_k\|_\infty \leq \varepsilon$. This leads to the following guarantee.

**Theorem 1** (Conditional Model-Relative Forward Invariance). *Given the learned dynamics model $\hat{F}$ and an error bound $\varepsilon$, if at every timestep $k$ the multi-step robust CBF problem defined by the constraints in equation 2 is feasible, and the true residual dynamics satisfy $\|w_k\|_\infty \leq \varepsilon$, then the state of the system $z_k$ is guaranteed to remain within the safe set $\mathcal{C}$ for all $k \geq 0$ (Blanchini, 1999).*

*Proof.* The proof is by construction and induction. At any state $z_k \in \mathcal{C}$, the feasibility of the QP in equation 3 implies the existence of a control sequence $\mathbf{u}$ that satisfies the robust multi-step CBF condition in equation 2. This condition, by its formulation, ensures that all intermediate states $z_{k+1}, \ldots, z_{k+H}$ remain within $\mathcal{C}$ for any possible realization of the error sequence where each $\|w_j\|_\infty \leq \varepsilon$. By applying the first action $u_0$ of this sequence, the resulting state $z_{k+1}$ is guaranteed to be in $\mathcal{C}$. The argument then applies recursively at timestep $k + 1$, as long as the condition stays feasible at $k + 1$ $\qquad\square$

While Theorem 1 establishes conditional recursive feasibility under the assumption that the QP remains feasible at every timestep, this requirement is standard but difficult to analytically guarantee in practical safe-control or safe-RL settings, especially when the dynamics model is learned. Prior model-based shielding and safe-exploration methods similarly rely on stepwise feasibility assumptions in their theoretical guarantees, while noting that infinite-horizon feasibility cannot be fully certified in practice and is instead supported empirically (Wang & Zhu, 2024; Anderson et al., 2023; Banerjee et al., 2024; Wachi et al., 2023). Consistent with this common limitation, we find that the QP in our framework is feasible in over 98% of timesteps, indicating that the theoretical assumption is well-satisfied in practice.

**Probabilistic Connection to the Physical System.** The deterministic guarantee of Theorem 1 is conditioned on the validity of the error bound $\varepsilon$. In practice, $\varepsilon$ is estimated empirically from a finite hold-out validation dataset, $D_{val}$, as the maximum observed one-step prediction error. The connection between this empirical bound and the true, underlying error distribution is necessarily probabilistic, but the bound is maintained with high probability. Theorem 2 formalizes this connection. The proof is given in Appendix A.2.

**Theorem 2** (High-Probability Model Accuracy). *Let $\epsilon_1, \ldots, \epsilon_N$ be a set of i.i.d. sampled model errors from our learned model $\hat{F}$. Assume that the probability of any two samples being equal is zero. Choose a quantile $0 < q < 1$ and let $\varepsilon$ be the $\lceil qN \rceil$'th smallest value among $\epsilon_1, \ldots, \epsilon_N$. Then*

$$Pr[\|F(s_k, u_k) - \hat{F}(s_k, u_k)\|_\infty > \varepsilon] \leq 1 - q + \frac{1}{(2N)^{1/3}} + \frac{1}{4(2^{1/3})N^{2/3}}.$$

**Corollary 1** (Probabilistic Forward Invariance over Finite Horizon). *Let $\delta = 1 - q + \frac{1}{(2N)^{1/3}} + \frac{1}{4(2^{1/3})N^{2/3}}$ be the failure probability of the empirical error bound $\epsilon$ from Theorem 2. If the multi-step robust CBF problem (Eq. 3) is feasible at every timestep $k \in \{0, \ldots, K - 1\}$ over a finite horizon of $K$ steps, then the true system state $z_k$ remains within the safe set $\mathcal{C}$ for all $k \in \{0, \ldots, K\}$ with probability $P \geq 1 - K\delta$.*

*Proof Sketch.* By Theorem 2, $\Pr(\|w_k\|_\infty \leq \epsilon) \geq 1 - \delta$ for each timestep $k$. By union bound over $K$ timesteps, $\Pr(\forall k \in \{0, \ldots, K - 1\} : \|w_k\|_\infty \leq \epsilon) \geq 1 - K\delta$. Conditioning Theorem 1's forward invariance on this high-probability event yields the result. $\qquad\square$

## 5 Experimental Evaluation

We conduct experiments to evaluate RAMPS on a suite of challenging control tasks. Our evaluation is designed to answer three primary research questions: **1. Safety Analysis**: Does RAMPS reduce safety violations more effectively than state-of-the-art safe RL algorithms? **2. Safety-Performance Tradeoff**: Does the minimally invasive nature of RAMPS allow the agent to learn a high-performing policy? **3. Role of Model Expressiveness** Does improved representational power of the learned dynamics model enhance the shielding performance of RAMPS?

**Environments.** We evaluate our method on five challenging environments. **Pendulum** is a classic low-dimensional control task. **SafeHopper**, **SafeCheetah**, **SafeAnt** and **SafeHumanoid** are high-dimensional locomotion tasks from the Safety-Gymnasium benchmark (Ji et al., 2023). **Safe-Humanoid** is a challenging benchmarks due to their high-dimensional state (348) and action spaces (17) and the complex, unstable dynamics of legged locomotion, where sophisticated coordination is required to prevent falling.

**Baselines.** We compare RAMPS against two classes of baselines. First, we consider state-of-the-art Constrained Markov Decision Process (CMDP) algorithms that optimize for reward while treating safety as a constraint: **PPOSaute** (Sootla et al., 2022a), **P3O** (Zhang et al., 2022), and **CUP** (Yang et al., 2022). We use the implementations from the OmniSafe-RL library (Ji et al., 2024). We compare against these methods because, unlike many symbolic approaches, they are capable of operating in the high-dimensional environments we consider. We discuss additional baselines in Appendix A.5.3.

Second, we compare against methods architecturally similar to RAMPS, which also learn a dynamics model for shielding. We selected **SPICE** (Anderson et al., 2023), which learns a simple linear model; we refer to this as **SPICE + L**. To provide a direct comparison of modeling techniques, we also implemented **SPICE + K**, a variant where we replace the original linear model with our learned Koopman operator. We found that while **SPICE + L** failed to scale to the high-dimensional SafeHopper and SafeCheetah environments, **SPICE + K** was able to produce a stable model. We attempted comparisons with other relevant MPS/MPC techniques - DMPS (Banerjee et al., 2024), VELM (Wang & Zhu, 2024), MASE (Wachi et al., 2023), and Conservative Safety Critics (Bharadhwaj et al., 2021b), but these methods failed to achieve stable training on the high-dimensional locomotion tasks, accumulating over 1000 violations within the first 20-30k environment interactions. More details are in Appendix A.5.3.

**Implementation Details.** To analyze the impact of the learned dynamics model, we evaluate two versions of our RAMPS framework: **RAMPS + L**, which uses a simple linear model learned via regularized regression in the original state space, and **RAMPS + K**, which uses the Deep Koopman Operator. The underlying policy for RAMPS variants is trained with PPO and SAC. For all baselines, we add a penalty reward of -100 and terminate the episode upon a safety violation to provide a clear learning signal. For the CMDP baselines, the cost is 1 for a violation and 0 otherwise. We also ran CMDP baselines using only the sparse violation cost (cost = 1 for a violation, 0 otherwise) without the -100 episode-termination penalty; under this protocol the CMDP baselines failed to learn a safe or performant policy within our training budget. For all experiments involving RAMPS and SPICE, we use a maximum prediction horizon of $H_{max} = 5$, which is justified in A.3. We use OSQP Stellato et al. (2020) to solve the quadratic program.

### 5.1 Results and Analysis

**Safety Analysis.** As detailed in Table 1, all variants of RAMPS demonstrates a substantial reduction in cumulative safety violations compared to various baselines across different environments. This effect is particularly pronounced in high-dimensional tasks like **SafeHopper**, **SafeCheetah**, **SafeAnt** and **SafeHumanoid**, where RAMPS variants typically exhibit significantly fewer violations than other methods. The violation curves in Figure 1, 6 visually reinforce these findings; while other methods often show a continued accumulation of violations during training, the curves for RAMPS variants tend to flatten much earlier in the training phase, indicating the shield's success in mitigating unsafe actions. This suggests that our multi-step shielding approach provides robust safety assurances, especially where optimization-based or other model-predictive methods may face challenges.

The effectiveness of RAMPS stems from its robust shielding framework, rather than solely relying on the underlying dynamics model's accuracy. A comparison between RAMPS + K and SPICE + K, both

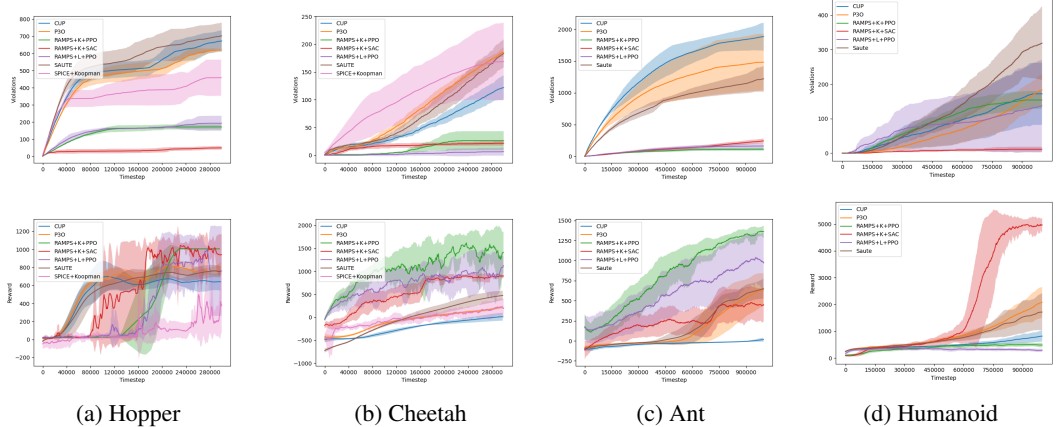

|                | (a) Hopper | (b) Cheetah | (c) Ant | (d) Humanoid |

Figure 1: Cumulative Safety violations (top row in each subfigure) and episodic reward (bottom row) for all high-dimensional environments.

Table 1: Cumulative safety violations during training. Failed indicates that training quit or the agent never completed a safe episode. L = Linear Regression baseline; K = Koopman Dynamics model.

| Algorithm | Pendulum | SafeHopper | SafeCheetah | SafeAnt | SafeHumanoid |
|---|---|---|---|---|---|
| SauteRL | $91 \pm 22$ | $703 \pm 78$ | $183 \pm 25$ | $1221 \pm 203$ | $319 \pm 106$ |
| CUP | $184 \pm 225$ | $673 \pm 63$ | $122 \pm 22$ | $1883 \pm 221$ | $172 \pm 90$ |
| P3O | $173 \pm 166$ | $620 \pm 6$ | $185 \pm 8$ | $1481 \pm 446$ | $183 \pm 45$ |
| SPICE + L | $495 \pm 128$ | Failed | Failed | Failed | Failed |
| SPICE + K | $87 \pm 8$ | $459 \pm 105$ | $169 \pm 70$ | Failed | Failed |
| RAMPS + L + PPO | $69 \pm 6$ | $193 \pm 44$ | $\mathbf{7 \pm 7}$ | $162 \pm 42$ | $137 \pm 134$ |
| RAMPS + K + PPO | $53 \pm 6$ | $172 \pm 15$ | $26 \pm 17$ | $\mathbf{111 \pm 23}$ | $154 \pm 25$ |
| RAMPS + K + SAC | $\mathbf{25 \pm 26}$ | $\mathbf{49 \pm 10}$ | $21 \pm 4$ | $242 \pm 38$ | $\mathbf{11 \pm 7}$ |

utilizing Koopman dynamics, reveals that RAMPS consistently achieves superior safety performance. This difference highlights RAMPS's ability to operate effectively even with an imperfect model, due to its explicit accounting for model error through robust multi-step predictions. While methods like SPICE typically require a highly accurate model with minimal error to ensure stable performance, RAMPS's design allows it to maintain safety guarantees across a broader range of model accuracies. Furthermore, the real-time operation of the RAMPS shield is highly efficient. As detailed in Table 2, the mean per-step computation time ranges from just 0.23 ms for Pendulum to 0.40 ms for the high-dimensional Ant environment, suggesting feasibility for real-time control loops.

**The Safety-Performance Tradeoff.** A critical aspect of safe RL is balancing stringent safety with high task performance. The reward curves in Figure 1 illustrate that RAMPS effectively navigates this tradeoff. Across environments, RAMPS achieves strong safety while obtaining competitive, and often superior, task rewards compared to the baselines. This suggests that the shield provides necessary interventions without being overly conservative, allowing the policy to exploit high-reward regions.

**Policy-Agnostic Shielding.** We evaluate RAMPS with both PPO (on-policy) and SAC (off-policy) to highlight that the shield operates independently of the underlying RL algorithm. SAC is generally more reliable, especially in high-dimensional settings such as **SafeHumanoid**, while PPO performs competitively and even surpasses SAC on **SafeAnt**. These results indicate that RAMPS is compatible with multiple learning paradigms and scales effectively to challenging continuous-control tasks.

The PPO instability observed on Humanoid is not a shield-specific failure but a known limitation of on-policy methods under action modification. Prior work shows that even simple invalid-action masking, structurally analogous to shielding because the executed action differs from the policy's proposal, can cause PPO's KL divergence to spike and training to collapse (Huang & Ontañón, 2020; Hou et al., 2023). Similar sensitivity has been documented in safe-RL algorithms such as CPO and

primal–dual CMDP methods, where constraint-induced distribution shift destabilizes updates without additional safeguards (Achiam et al., 2017; Paternain et al., 2019; Ding et al., 2020). In contrast, off-policy approaches exhibit greater robustness to distribution mismatch (Liu et al., 2022).

**Empirical Feasibility of Multi-step constraints** A crucial element of our framework's reliability is the practical feasibility of the multi-step QP. We analyzed the action selection distribution, and the results (detailed in Appendix A.5.5, Table 3) confirm our shield is highly robust. For the complex locomotion tasks, the backup policy was invoked in **less than 2% of all timesteps**, and for Pendulum and SafeHumanoid, it was never used at all. This demonstrates that our primary shield consistently finds a feasible, safe solution, validating the empirical stability of our approach and showing that the conditional guarantee of Theorem 1 is almost always active.

**Role of Model Expressiveness**. The choice of dynamics model within RAMPS can influence this balance between safety and reward, particularly in environments with complex dynamics. While both RAMPS + L (simple linear model) and RAMPS + K (Koopman model) offer significant safety improvements, the more expressive Koopman model generally supports better reward performance. This is observed in all environments, but particularly in **SafeCheetah**, where RAMPS + L achieves extremely low violations but shows lower reward accumulation compared to RAMPS + K. As shown in Appendix A.6, this is an outcome of the simpler linear model leading to a more conservative shield (due to larger estimated error bounds), resulting in interventions with larger deviations from the neural action. This hinder the agent's ability to learn a policy that maximizes the reward. In contrast, the more accurate Koopman model allows for a less conservative, yet still provably safe, shield, thereby improving the overall safety-performance balance.

**Ablation Analysis.** We performed an extensive ablation analysis, detailed in Appendix A.3, to validate the design principles of the RAMPS framework. These studies confirm that robust, multi-step shielding is a co-designed system requiring a careful balance of competing factors. Our most critical finding is that explicit **robustness to model error is the essential component for safety**; removing the error-aware tightening term proved catastrophic, leading to continuous safety violations regardless of other hyperparameter settings (Figure 2).

Furthermore, the ablations justify our hyperparameter choices by exploring key trade-offs. The prediction horizon $H$ must be long enough to resolve high relative-degree traps but short enough to avoid compounding model error (Figure 4). The CBF decay rate $\lambda$ must be permissive enough to ensure the underlying QP remains feasible, as an overly conservative setting harms both safety and reward (Figure 3). Finally, we show that a high-confidence error bound (99th percentile) is a prerequisite for achieving both safety and high reward, as it creates a more stable learning environment (Figure 5). We further evaluate RAMPS under *multi-dimensional safety constraints* to demonstrate scalability A.7. In the SAFEHUMANOID benchmark, we simultaneously constrain the 3 coordinate and 18 joint angular velocities (a 21-dimensional safety set). RAMPS accumulates only **256 violations**, whereas CMDP-based baselines exceed **3000 violations** and fail to learn a safe policy, reflected by their steadily increasing violation curves. Additonally, RAMPS is the only method that attains a high task reward of **5,000**, while CMDP baselines plateau near **500**. Together, these results show that RAMPS maintains safety even under high-dimensional constraints without sacrificing performance. Collectively, these results validate our methodology and demonstrate that effective shielding arises from a calibrated synthesis of all framework components.

## 6 CONCLUSION

We present RAMPS, a scalable model-predictive shielding framework that enables safe policy learning for complex, high-dimensional systems. The core of our approach is the synergy between a learned, linear dynamics model and a robust, multi-step safety shield. By leveraging a linear representation, which can range from a simple regression to a more complex latent model like the koopman operator, RAMPS remains computationally tractable. Its multi-step, adaptive-horizon Control Barrier Function provides strong foresight to prevent safety violations, even when the learned model is an imperfect approximation of the true dynamics. Experiments on a suite of challenging environments demonstrate the efficacy of RAMPS, showing it can dramatically reduce safety violations while maintaining high task performance. Its ability to learn a reliable safety model from a few samples makes it particularly well-suited for deployment of reinforcement learning agents in safety-critical applications.

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
