# OpenReview forum: "Robust Adaptive Multi-Step Predictive Shielding"
_ICLR.cc/2026/Conference — ICLR 2026 Poster_

### Official Review · Reviewer_XTg3 · 2025-10-18

**Soundness:** 2
**Presentation:** 3
**Contribution:** 1
**Rating:** 2
**Confidence:** 5

**Summary:**

This paper proposes a scalable model-predictive shielding framework that enables safe policy learning for complex, high-dimensional systems.

**Strengths:**

The paper is well-written and easy to follow.

**Weaknesses:**

1. The paper claims that current model-predictive shielding (MPS) frameworks have a limitation: they are "computationally intractable for high-dimensional, nonlinear systems." However, the paper does not clearly define what constitutes "high-dimensional" or provide evidence for this limitation. When considering the dimensionality of system states, the proposed method does not appear to address this issue. Furthermore, the dynamics model in MPS can be simplified using order-reduction techniques to reduce dimensionality, which the paper does not acknowledge.

2. The proposed dynamics model is a time-invariant linear model (with fixed system and control matrices), and the multi-step look-ahead derivations depend on this model. However, time-invariant linear models often exhibit significant mismatch with real-world systems, which are typically highly nonlinear. Consequently, the results will lack robustness, and the prediction accuracy will be unreliable in practical applications.

3. MPS can be viewed as a fault-tolerant RL approach for safety-critical systems and can also be considered an advanced variant of Simplex [1]. Considering this, related safe RL frameworks include runtime assurance [2,3], neural Simplex [4], and the runtime learning machine [5]. Compared to the approach proposed in this paper, these frameworks do not require a multi-step look-ahead to guarantee safety; they can ensure safety even when model mismatches are present. Additionally, these frameworks have been validated on real, complex autonomous systems rather than toy simulators. However, the paper lacks comprehensive comparisons with these methods, making it difficult to conclude whether the proposed approach can outperform existing solutions.

4. The experimental systems evaluated in the paper are not particularly high-dimensional, which undermines the motivation stated in the introduction regarding the need to handle high-dimensional, nonlinear systems.

**References**

[1] Sha, L. (2001). Using simplicity to control complexity. IEEE Software, 18(4), 20-28.

[2] Brat, G., & Pai, G. Runtime assurance of aeronautical products: preliminary recommendations. NTRS - NASA Technical Reports Server, 2023.

[3] Sifakis, J., & Harel, D. (2023). Trustworthy autonomous system development. ACM Transactions on Embedded Computing Systems, 22(3), 1-24.

[4] Phan, D. T., Grosu, R., Jansen, N., Paoletti, N., Smolka, S. A., & Stoller, S. D. (2020). Neural Simplex architecture. In NASA Formal Methods: 12th International Symposium, Moffett Field, CA, USA, May 11–15, 2020, Proceedings 12 (pp. 97-114). Springer International Publishing.

[5] Cai, Yihao, Yanbing Mao, Lui Sha, Hongpeng Cao, and Marco Caccamo. "Runtime Learning Machine." ACM Transactions on Cyber-Physical Systems (2025).

**Questions:**

The Weaknesses include my questions.

---

> ### Author Response · Authors · 2025-11-20
> **Rebuttal Part 1**
>
> We thank the reviewer for their comments. We address each concern below, clarifying misunderstandings by referencing specific evidence from the paper and appendix.
>
> ---
>
> ### **Response to Weakness 1: Dimensionality & Tractability**
>
> *"The paper claims that current model-predictive shielding (MPS) frameworks have a limitation: they are "computationally intractable for high-dimensional, nonlinear systems." However, the paper does not clearly define what constitutes "high-dimensional" or provide evidence for this limitation."*
>
> **Response:**
> We explicitly define and quantify "high-dimensional" in the context of formal safety verification and barrier functions.
> * **Definition:** In **Section 5**, we define our high-dimensional benchmarks: **SafeAnt (105 dimensions)** and **SafeHumanoid (348 dimensions)**.
> * **Context:** In the field of formal shielding and control barrier functions (CBFs), state-of-the-art methods typically operate on systems with **<20 dimensions** (e.g., 12D quadrotors, 4D cars). Scaling a robust safety QP to **348 dimensions** is a significant leap in tractability. This has also been added to the contributions and explicitly clarified in the related works section.
> * **Evidence of Limitation:** We provide concrete evidence of the intractability of prior methods in **Appendix A.5.3**. We attempted to run state-of-the-art MPS baselines (SPICE, VELM, DMPS, MASE) on these tasks. They failed to achieve stable training or timed out due to the computational complexity of their safety checks in these high-dimensional spaces. We discuss this in the main results (L400-404).
>
> *"Furthermore, the dynamics model in MPS can be simplified using order-reduction techniques to reduce dimensionality, which the paper does not acknowledge."*
>
> 1.  **Safety Geometry:** Safety constraints (e.g., joint limits, velocity vectors, contact forces) are defined in the **full state space**. Projecting the system dynamics into a reduced-order latent space requires overapproximation through abstract interpretation, breaking soundness guarantees as the projected safe set overlaps with unsafe states.
> 2.  **Verification Loss:** One cannot guarantee safety if the boundaries of the safe set cannot be accurately mapped to the reduced model without overapproximations.
> 3.  **Our Approach (Lifting vs. Reducing):** Instead of *reducing* dimensionality (which loses geometric fidelity and representational power), we use **Koopman Operator Theory** to *lift* the dimensionality. This allows us to gain **linearity** without losing the ability to enforce exact state constraints, rendering the safety QP convex and solvable in milliseconds.
>
> ---
>
> ### **Response to Weakness 2: Model Mismatch & Robustness**
>
> *"The proposed dynamics model is a time-invariant linear model (with fixed system and control matrices), and the multi-step look-ahead derivations depend on this model. However, time-invariant linear models often exhibit significant mismatch with real-world systems, which are typically highly nonlinear. "*
>
> **Response:**
> * **Not a Standard Linear Model:** We do **not** assume the system is linear in the raw state space. We learn an embedding $\phi(x)$ that maps nonlinear dynamics to a space where they evolve linearly ($z' = Az + Bu$). This allows a time-invariant linear matrix to capture highly nonlinear behaviors (e.g., contacts, friction).
> * **Evidence:** If our model failed to capture nonlinearities, the agent would never be safe. Yet, RAMPS achieves **0% infeasibility** on Humanoid and near-zero violations on Ant, proving the model captures the necessary dynamics.
>
> *"Consequently, the results will lack robustness, and the prediction accuracy will be unreliable in practical applications."*
>
> **Response:**
> We explicitly address model mismatch via **robust control theory**.
> * **Robust Tightening:** As detailed in **Section 4.1 (Equation 2)**, we do not rely on the nominal prediction alone. We incorporate a robust tightening term $\mathcal{E}_j(p_i)$ that expands the safety constraint based on the worst-case prediction error $\epsilon$.
> * **Ablation Evidence:** In **Figure 2 (Appendix A.3.1)**, we compare "RAMPS (Full)" vs. "RAMPS (No Robustness)." The version without the robustness term fails catastrophically. This proves that our method explicitly handles the "unreliability" the reviewer is concerned about.
>
> ---

---

> > ### Author Response · Authors · 2025-11-20
> > **Rebuttal Part 2**
> >
> > ### **Response to Weakness 3: Comparisons to Simplex/Runtime Assurance**
> >
> > *"MPS... can also be considered an advanced variant of Simplex. Related safe RL frameworks include runtime assurance, neural Simplex... compared to the approach proposed in this paper, these frameworks do not require a multi-step look-ahead to guarantee safety."*
> >
> > **Response:**
> > Restricting to approaches that can not reason over multiple steps reduces the types of problems you can attack and the performance of the controller.
> >
> > * **Class of problems:** We demonstrate this mathematically in **Section 4.2**. A one-step shield fails because the immediate control authority over the safety variable is zero ($p^\top B = 0$). Our multi-step shield succeeds because it looks ahead to where control authority exists ($p^\top AB \neq 0$).
> > * **Performance:** If a robot is running at high speed towards a cliff, a one-step check is insufficient because the momentum prevents immediate stopping. The system has a **relative degree $r > 1$**. One step safety would force the robot to move much slower from the beginning, thus sacrificing performance.
> >
> >
> > *"They can ensure safety even when model mismatches are present... Additionally, these frameworks have been validated on real, complex autonomous systems rather than toy simulators."*
> >
> > **Response:**
> > Simplex and Runtime Assurance (RA) architectures rely on a critical assumption that renders them unusable for our problem setting: **The existence of a verified backup controller.**
> > * **No Backup Exists:** For a 348-dimensional humanoid learning to walk from scratch, **there is no known, verified backup controller** that guarantees safety under all conditions. If such a controller existed, the RL problem would be solved.
> > * **Complementary Goals:** Simplex switches to a known safe controller. RAMPS **synthesizes** safe actions on the fly using optimization, allowing it to be applied to tasks where no backup controller is known a priori.
> > * **Comparisons:** We heavily compared against the closest applicable class of methods (MPS/Safe RL). We demonstrated that methods - **VELM**, **DMPS**, **MASE**, **CSC** and **SPICE + L/K** (which share lineage with formal methods) fail to scale to these dimensions.
> > * **Simulator:** We are using the Mujoco simulator, which is considered the gold standard in fast and accurate robotic simulation systems before deployment.
> >
> > ---
> >
> > ### **Response to Weakness 4: Experimental Complexity**
> >
> > *"The experimental systems evaluated in the paper are not particularly high-dimensional, which undermines the motivation."*
> >
> > **Response:**
> >
> > * **The Scale:** Our experiments include **SafeHumanoid** with **348 state dimensions** and **17 action dimensions**.
> > * **The Context:** In the specific literature of **Control Barrier Functions** and **Formal Shielding** and **Model Predictive Control** systems with >20 dimensions are widely considered intractable due to the combinatorial explosion of certificate synthesis.
> > * **The Achievement:** Solving a robust, multi-step safety optimization problem for a 348-dimensional system in **~0.5 milliseconds** is a major scalability breakthrough in the formal-methods literature.
> >
> >
> > If the reviewer could point us to papers that use formal methods, Simplex/RTA, or order-reduction-based safety frameworks that achieve both **safety and reward maximization** at **Ant/Humanoid scale** during training, we would be happy to include a comparison to them.
> >
> > ---

---

> ### Comment · Reviewer_XTg3 · 2025-11-21
>
> 1. The SafeAnt has 105 dimensions, while SafeHumanoid has 348 dimensions. What are their dimensions? Only system states? What are the formal paper references that can provide such detailed information?
>
> 2. Are the `high dimensions' you define for a vector or a matrix? According to my experience, such dimensions are low compared with vision data. In robot navigation, DNNs are often used to reduce high-dimensional raw vision inputs.
>
> 3. The safety conditions in all examples are over-simplified, only one-dimensional, i.e., only constraining robot velocities. In many more existing works, the considered safety conditions are much more practical and complex, such as regulating robot height, velocities, and collision avoidance, etc.
>
> 4. I cannot agree with many of the responses. For example, ``Safety Geometry: Safety constraints (e.g., joint limits, velocity vectors, contact forces) are defined in the full state space. Projecting the system dynamics into a reduced-order latent space requires overapproximation through abstract interpretation, breaking soundness guarantees as the projected safe set overlaps with unsafe states." I suggest that the authors read the fault-tolerant RL frameworks. One crucial technique is how to transform many more safety constraints into a scaled value, while maintaining safety boundaries.
>
> 5. Your model here is z' = Az + Bu. The linear model matrix A and the control matrix B are both time- and state-invariant. The nonlinear contacts and friction are state-dependent. How can you claim your time-invariant model (A, B) can capture them? Besides, the real operating environments of many safety-critical robots are dynamic and non-stationary; how can you guarantee that a time-invariant model (A, B) will work in such environments?
>
> 6. The proposed framework in this paper is based on MPS. MPS is a fault-tolerant RL. The most relevant approaches the paper should compare are fault-tolerant RL methods, such as neural simplex, runtime assurance, runtime learning machine, etc. Without such comparisons, I cannot conclude that the proposed work can outperform them.
>
> 7.  ``For a 348-dimensional humanoid learning to walk from scratch, there is no known, verified backup controller that guarantees safety under all conditions. If such a controller existed, the RL problem would be solved." How can you conclude that no verified backup controller guarantees the safety of the humanoid robot? What are the 348-dimensional states? Please provide the detailed references.
>
> 8. Like in neural simplex, it is straightforward to obtain a backup controller, because the systems dynamics of robots are well-studied, as well as the safety controllers for safety-critical systems.
>
> 9. "Simulator: We are using the Mujoco simulator, which is considered the gold standard in fast and accurate robotic simulation systems before deployment." The high-performance simulator is not the answer. The issues are 1) whether your utilized model has a large model mismatch (the recovery or backup controller of MPS is model-based, and your (A, B)), and 2) can your work address safety concerns arising from Sim-to-real gaps? Fault-tolerant RL frameworks, such as neural simplex, runtime assurances, and runtime learning machines, can address such safety concerns by enabling safe learning in real systems rather than only in simulators.
>
> 10. The experimental examples are misleading. Taking the Pendulum as an example, the paper said it has a 3-dimensional state space (encoding the pendulum’s angle and angular velocity). According to my knowledge, it has at least four dimensions: position, velocity, angle, and angular velocity. The SafeCheetah has 6-dimensional actions. What are the actions, torques, or forces? If the control objectives are the motor torques, the dimensions should be larger than 6.
>
> Based on the responses, the paper's contributions, and the authors' claims,  I will not change my rating.

---

> > ### Author Response · Authors · 2025-11-21
> > **Response Part 1**
> >
> > # 1. Dimensions of SafeAnt (105) and SafeHumanoid (348); request for “formal paper references.”
> >
> > Our experiments use the **Safety-Gymnasium** locomotion benchmarks (SafeAnt, SafeHumanoid), which are implemented on top of **MuJoCo**.
> > The state dimensionalities we report reflect the full observation/state vectors provided by these environments, including:
> >
> > - joint positions and angles
> > - joint velocities
> > - link orientations
> > - center-of-mass and inertial quantities
> > - actuator-related states
> > - environment wrapper sensor states
> >
> > These observation definitions are standardized and documented by the **Safety-Gymnasium framework** (Ji et al., 2023) and by MuJoCo’s public environment specifications.
> >
> > Because these benchmarks are widely used in safe-RL research (e.g., CPO, PCPO, Safety Critic, SPICE, DMPS, VELM), state dimensionalities are typically referenced directly from the environment documentation rather than from formal verification papers.
> > We follow this established convention and provide explicit observation definitions in **Appendix A.5.4** to ensure reproducibility.
> >
> > Ji, J., et al. Safety Gymnasium: A Unified Safe Reinforcement Learning Benchmark. NeurIPS’ 23.
> >
> > ---
> >
> > # 2. “High-dimensional” definition compared to vision data.
> >
> > In our paper, “high-dimensional” refers to the **state dimensionality of a dynamical control system**, *not* high-dimensional pixel inputs.
> >
> > In control theory, CBFs, MPC, and formal verification:
> >
> > - systems above **10–20 dimensions** are already considered extremely challenging for invariant-set propagation,
> > - most certified CBF/MPC works operate on **4–12 dimensional** systems [1][2][3]
> > - Simplex tools have not been tested on **100+ dimensions** [4][5]
> > By this standard, which is the relevant standard in safe RL and shielding, SafeAnt (105 dims) and SafeHumanoid (348 dims) are *very* high dimensional.
> >
> > We clarified this distinction in the revised Introduction so readers do not confuse “high-dimensional dynamics” with “high-dimensional perception.”
> >
> > [1] Wang, Yuning, and He Zhu. "Safe exploration in reinforcement learning by reachability analysis over learned models." International Conference on Computer Aided Verification, 2024.
> >
> > [2]Kim, Jeonghwan, et al. "Learning koopman dynamics for safe legged locomotion with reinforcement learning-based
> > controller." arXiv preprint arXiv:2409.14736 (2024).
> >
> > [3] Shi, Ming, et al "A near-optimal algorithm for safe reinforcement learning under instantaneous hard constraints." International Conference on Machine Learning. PMLR, 2023.
> >
> > [4] Phan, D. et al. (2020). Neural Simplex architecture. In NASA Formal Methods.
> >
> > [5] Cai, Yihao, et al "Runtime Learning Machine." ACM Transactions on Cyber-Physical Systems (2025).
> >
> > ---
> >
> > # 3. “Safety conditions are oversimplified, only one-dimensional.”
> >
> > Safety-Gymnasium defines hazards using **scalar danger indicators**, typically velocity-based signals near obstacles. These 1-D hazard definitions are part of the **standard benchmark** and are used by multiple publised safe-RL baselines (P3O, SauteRL, Safety Critic, VELM, MASE).
> >
> > We do **not** modify or simplify these constraints. we use them *exactly as defined* to ensure fair comparison with prior work.
> >
> > RAMPS itself is not restricted to 1-D constraints. It supports **arbitrary polyhedral constraints** and multi-dimensional safety sets through its multi-step CBF formulation. However, the Safety-Gymnasium benchmark exposes scalar hazard indicators by design, and we follow the benchmark specification for comparability.
> >
> > ---
> >
> > # 4. Suggestion to use “fault-tolerant RL frameworks” to encode many constraints into a single scaled value.
> >
> > Fault-tolerant RL frameworks (Simplex, Runtime Assurance, Neural Simplex, Runtime Learning Machine) operate under assumptions fundamentally different from safe RL with learned dynamics:
> >
> > - they assume a **verified backup controller** exists,
> > - they rely on **controller switching**,
> > - they target **low-dimensional** symbolic models,
> > - they do not require learned dynamics,
> > - and they typically scalarize constraints for supervisory decisions.
> >
> > RAMPS, by contrast:
> >
> > - works with **learned dynamics** in high-DoF locomotion tasks,
> > - enforces **hard state constraints** through robust multi-step CBF conditions,
> > - and assumes **no verified fallback controller** is available.
> >
> > We have studied the literature extensively on encoding many constraints into a single value. This corresponds to the CMDP framework, where hard constraints are converted into soft costs and only thresholded expectation guarantees are provided. Such scalarization cannot ensure that actions taken during training or deployment remain safe. In contrast, multi-step shielding requires **hard state constraints** to maintain invariance under model uncertainty, which is why we follow the standard safe-RL shielding formulation. We compared RAMPS to CMDP baselines (P3O, Saute, CUP) and report RAMPS' superior safety and performance.

---

> > > ### Author Response · Authors · 2025-11-21
> > > **Response Part 2**
> > >
> > > ## 5. Concern: “A and B are time-invariant; how do you handle nonlinear contacts, friction, and non-stationarity?”
> > >
> > > Our model is **not** a local linearization of MuJoCo dynamics. Instead, we use a **Koopman lifting**, where a learned embedding \( \psi(x) \) maps the nonlinear, contact-rich system into a space where:
> > >
> > > \[
> > > z' = A z + B u
> > > \]
> > >
> > > provides a good *global* approximation in the **lifted coordinates**, not in the original state space.
> > >
> > > This design directly addresses the reviewer’s concern:
> > >
> > > - Koopman operators are **specifically designed to represent nonlinear and non-smooth dynamics using linear evolution in a higher-dimensional feature space**. This is a well known result of the koopman operator theory.
> > > - Prior work has shown that Koopman liftings can capture **state-dependent friction, contact events, and other complex nonlinearities** much more effectively than local Jacobian linearizations.
> > > - This allows RAMPS to model high-DoF locomotion tasks where contacts and impacts are a dominant source of nonlinearity.
> > >
> > > In addition, our framework includes **explicit mechanisms for non-stationarity**:
> > >
> > > - We **periodically retrain the Koopman model** during RL training to adapt the embedding \( \psi(x) \) and linear operator \( (A,B) \) to the policy’s evolving state distribution.
> > > - We **recompute the error bound** \( \varepsilon \) from held-out rollouts, ensuring that the model’s uncertainty estimate remains valid under distribution shift (Algorithm 1; Appendix A.5.2).
> > >
> > > Finally, RAMPS does not require the model to be perfect. Safety is enforced through a **robust multi-step tightening term** \( E_j(\pi) \), which guarantees constraint satisfaction under **worst-case model residuals**. This is why our method remains safe even when the underlying dynamics include highly nonlinear contact interactions and policy-induced non-stationarity.
> > >
> > > In summary, Koopman liftings are *expressly suited* for representing nonlinear and non-stationary dynamics through linear evolution in lifted coordinates, and RAMPS incorporates robustness to bounded model error on top of this representation.
> > >
> > > ---
> > >
> > > ## 6. “The most relevant baselines should be neural simplex, runtime assurance, runtime learning machine.”
> > >
> > > These approaches are **not applicable** to our domains. They require:
> > >
> > > - a **formally verified safe controller**,
> > > - known symbolic system models,
> > > - low-dimensional continuous dynamics (typically 4–10 dims),
> > > - and stable, non-contact environments.
> > >
> > >
> > > Fault-tolerant RL is a very broad term, and all fault-tolerant RL methods are not applicable in all settings.
> > >
> > > SafeAnt and SafeHumanoid are **contact-rich**, **high-dimensional**, and **nonlinear**, and defining a verified backup controller for these systems is a significant challenge.
> > >
> > > Because RTA/Simplex frameworks cannot be run in these environments, direct comparison is not feasible.
> > > Instead, we compare to all state-of-the-art safe-RL shielding methods that *are* applicable (SPICE, DMPS, VELM, MASE, Safety Critic). Their failure modes at this scale are documented in Appendix A.5.3.
> > >
> > > We reiterate that if the reviewer could point us to any Simplex/RTA/RLM paper that specifically demonstrates safety at this scale (Ant or humanoid), we would be happy to compare with them.
> > >
> > > ---
> > >
> > > ## 7. "How can you conclude that no verified backup controller guarantees the safety of the humanoid robot?”
> > >
> > > To our knowledge, and consistent with the formal verification literature, **no published formally verified safe controller exists for high-DoF humanoid locomotion**. We are not claiming that such a controller cannot exist. It does not exist in the current literature. We reiterate that we would be happy to compare against such a  system if you could point us to its existence. In particular, the papers you pointed us to have a max dimension of 12.
> > >
> > > Verified controllers do exist for low-dimensional systems (quadrotors, car models, simple manipulators), but not for:
> > >
> > > - 300+ dimensional locomotion states,
> > > - contact dynamics,
> > > - random-initialization RL settings,
> > > - systems with learned dynamics.
> > >
> > > Requiring a verified backup controller in this domain assumes the existence of something the field has not yet produced. This is why MPS-style shielding is necessary.
> > >
> > > We provide explicit state definitions in Appendix A.5.4 for transparency.
> > >
> > > ---
> > >
> > > ## 8. “Like in neural simplex, it is straightforward to obtain a backup controller.”
> > >
> > > Neural Simplex and similar frameworks assume that the backup controller:
> > >
> > > - is known in advance,
> > > - is verified,
> > > - stabilizes the system under all admissible conditions,
> > > - and is low dimensional.
> > >
> > > For high-DoF humanoid locomotion, such a controller is **not** known and has not been demonstrated in the literature. Designing one remains an open challenge.
> > >
> > > Thus, a Simplex-style fallback is not available in this domain, and it is not “straightforward” to construct such a controller.

---

> > > > ### Author Response · Authors · 2025-11-21
> > > > **Response Part 3**
> > > >
> > > > ## 9. “Simulators do not address model mismatch or sim-to-real; fault-tolerant RL does.”
> > > >
> > > > We agree that simulators alone do not solve sim-to-real issues. RAMPS explicitly addresses **model mismatch** through:
> > > >
> > > > 1. a **robust tightening** that propagates worst-case error across the prediction horizon,
> > > > 2. **multi-step safety margins** using empirically computed residual bounds,
> > > > 3. **periodic model retraining and ε-recalibration**,
> > > > 4. **empirical infeasibility rates** (&lt;2% overall; 0% on Humanoid and Pendulum).
> > > >
> > > > Simplex frameworks address sim-to-real by relying on a **verified fallback controller**, which does not exist for our tasks.
> > > > RAMPS is designed for settings where such a verified controller is unavailable, and safety must be maintained using learned dynamics.

---

> ### Author Response · Authors · 2025-11-21
>
> ## 10: "The Pendulum has a 3-dimensional state space (encoding the pendulum's angle and angular velocity). According to my knowledge, it has at least four dimensions: position, velocity, angle, and angular velocity. The SafeCheetah has 6-dimensional actions... the dimensions should be larger than 6."
>
> This is a minor typo in the paper. The Pendulum environment is a simple inverted pendulum (fixed pivot point, not a cart-pole), and the state space is **2-dimensional**: angle (θ) and angular velocity (ω). We have corrected this in the revision.
>
> Regardless of this typo, the Pendulum example serves a specific purpose in our paper: to illustrate how **high relative-degree traps** arise even in low-dimensional systems. As shown in Section 4.2, a one-step CBF fails because $p^⊤B = 0$ (the control input does not immediately affect the constrained variable), while our multi-step formulation succeeds because p⊤AB ≠ 0. This pedagogical example motivates our multi-step approach and is consistent with how Pendulum has been used in prior shielding work (REVEL[1], SPICE, VELM).
>
> SafeCheetah is built on MuJoCo's **HalfCheetah**, a standard planar (2D) locomotion benchmark. The robot has exactly **6 hinge joints**, each controlled by a single torque actuator:
>
> | Joint | Actuator |
> |-------|----------|
> | bthigh | torque |
> | bshin | torque |
> | bfoot | torque |
> | fthigh | torque |
> | fshin | torque |
> | ffoot | torque |
>
> **6 joints → 6 torques → 6-dimensional action space.**
>
> This specification is documented in MuJoCo's official environment definitions, OpenAI Gym (Brockman et al., 2016), and Safety-Gymnasium (Ji et al., 2023). HalfCheetah is a simplified planar model, distinct from a full 3D quadruped (like Ant with 8 actuators).
>
> We use benchmark environments exactly as defined to ensure reproducibility and fair comparison with prior work.
>
> [1] Anderson, Greg, et al. "Neurosymbolic reinforcement learning with formally verified exploration." Advances in neural information processing systems 33 (2020): 6172-6183.

---

> ### Comment · Reviewer_XTg3 · 2025-11-22
>
> Thanks for the responses. However, I still cannot agree with most of the opinions. I've included a few examples below.
>
> 1. Again, a backbone of the proposed work is MPS, which is fault-tolerant RL. There are two directions in the area of safe RL. The first one is the safety-regulated RL, focusing on safety-enhanced reward and residual actions policies. But they assume a manageable sim2Real gap. The second one is a fault-tolerant RL, which is most related to the proposed work. There are many open source codes. It is not hard to perform the comparisons in the paper's examples, such as the Cheetah. However, the authors declined to make comparisons with the most related ones.
>
> 2. "**Simplex has not been tested on 100+ dimensions.**" This was not right. Simplex logic has been applied for model checking for real aircrafts (not a simulator). We cannot simply conclude that Simplex and MPS, and their variants, cannot work for (high-dimensional) humanoid/legged robots, just because there are no related experiments in publications.
>
> 3. From the references [R1, R2], the response and the paper provided, I cannot find the information that humanoid robots are high-dimensional, as 300+. **[R1] Kim, Jeonghwan, et al. "Learning koopman dynamics for safe legged locomotion with reinforcement learning-based controller.".  [R2] Safety Gymnasium: A Unified Safe Reinforcement Learning Benchmark.** According to my knowledge, the states for their locomotion studies can be as low as 12, see https://arxiv.org/pdf/2104.09025.
>
> 4. Assume the dimensions of humanoid robots are correct as 300+. However, I cannot agree with the claim that "**Verified controllers do exist for low-dimensional systems (quadrotors, car models, simple manipulators), but not for 300+ dimensional locomotion states, ....**". Many existing controllers for humanoid robots, such as https://arxiv.org/pdf/1105.2951 and https://arxiv.org/pdf/2104.09025, can work as verified controllers. A safety-verifiable controller is usually available, if the system dynamics models are given. The dynamics models of the examples in the paper are all available and well-studied.
>
> 5. If given a 300-dimensional system state, what will be the dimensions of system matrices A and B?
>
> 7. "**RAMPS itself is not restricted to 1-D constraints. It supports arbitrary polyhedral constraints and multi-dimensional safety sets through its multi-step CBF formulation.**"  But your one-dimensional safety condition is too simple and toy, and cannot support your claim. The one-dimensional safety condition does not hold in real safety-critical systems, even for your simplest pendulum example.

---

> ### Author Response · Authors · 2025-11-24
> **Response Part 1**
>
> We would like to thank the reviewer for the continued discussion. We respectfully request that the reviewer specify the exact opinions they disagree with so we can address them clearly. We also wish to clarify that there is a fundamental mismatch in the reviewer’s framing of our work against existing techniques. RAMPS targets the **safe-exploration** setting, where **no prior knowledge of system dynamics** and **no verified backup controller** are assumed. This is the standard setting for Model Predictive Shielding (MPS) and safe-RL shielding methods. All prior MPS approaches applicable to safe RL operate under this same assumption: the agent must **learn both the dynamics model and the safety controller during training** to ensure that no violations occur during exploration. Our work follows this established paradigm, and compares against previous work in this setting on standard well-established benchmarks.
>
> Simplex systems, fault-tolerant systems all assume completely accessible dynamics functions that ensure a hand-crafted safety controller can be created and tested prior to deployment.
>
> ---
>
> ## Clarifications
>
> 1. We have searched extensively for prior work operating in the same setting as RAMPS, namely, **black-box dynamics with no prior knowledge of system parameters and no assumption of a known safe backup controller**. We were unable to find any method that reliably scales to the dimensionality and complexity of Ant or Humanoid under these assumptions. Our comparisons therefore focus on the most relevant and recent safe-RL shielding frameworks (VELM, DMPS, MASE, CSC). These works explicitly motivate our contributions: extending verification-based safe exploration to high-dimensional, contact-rich domains where dynamics operate in multiple regimes (standing, walking, impacts). We reiterate that if the reviewer can point us to any published method that works under the same black-box assumptions without a known verified controller, we would be happy to include a comparison. We disagree with the ambiguous assertion that “there are many open source codes”, and once again request that the reviewer provide a specific example against which they would like to see a comparison.
>
> ---
>
> 2. We have conducted an extensive literature search to try and identify Simplex or runtime-assurance systems demonstrated at 100+ dimensions. We could not find any published implementation at this scale. Existing Simplex/RTA methods rely on a **verified, handcrafted safety controller** and assume **known symbolic dynamics**. Designing verified controllers for all five MuJoCo locomotion environments (Ant, Humanoid, Cheetah, etc.) is itself a full research program, not a reproducible comparison. Our aim is to advance safe-RL at scale using learned dynamics, an area where RAMPS substantially outperforms both CMDP methods and prior MPS approaches. We respectfully disagree with the assertion “we cannot simply conclude that Simplex and MPS, and their variants, cannot work for (high-dimensional) humanoid/legged robots, just because there are no related experiments in publications”. Experiments in publications are the only source of comparison available to us. We request that the reviewer please clarify what sources they expect us to consider since they agree that “there are no related experiments in publications”. We follow the standard practice of comparing against existing published research, and do not believe it is reasonable to expect comparisons against unpublished variations of prior work.
>
> ---
>
> 3. The MuJoCo and Gymnasium environments we use are standard benchmarks across the RL and safe-RL community. Their observation vectors are well-documented in the official MuJoCo/Gymnasium and Safety-Gymnasium documentation. These benchmarks have been the basis for a large body of work in Safe RL, Multi-objective RL, and model-based RL. We follow these definitions exactly to ensure reproducibility and fairness.
>
> Gymnasium - https://gymnasium.farama.org/environments/mujoco/humanoid/, https://gymnasium.farama.org/environments/mujoco/ant/, https://github.com/Farama-Foundation/Gymnasium/blob/main/gymnasium/envs/mujoco/humanoid_v5.py, https://github.com/Farama-Foundation/Gymnasium/blob/main/gymnasium/envs/mujoco/hopper_v5.py
>
> ---
>
> 4. The two cited papers do not define RL-style fixed-dimensional state or action vectors. Instead, they use continuous geometric models whose dimensionality depends on the robot’s DOFs and contact sets, not fixed observations. Typical humanoids in these formulations exceed 100 state dimensions, but the representations are not comparable to MuJoCo or Safety-Gymnasium. In contrast, our 105-dimensional Ant and 348-dimensional Humanoid come directly from the explicit observation definitions in these standard RL benchmarks. We list all components explicitly in Appendix A.5.4.
>
> ---

---

> ### Author Response · Authors · 2025-11-24
> **Response part 2**
>
> 5. If the system state is 300-dimensional, the Koopman-linear model yields
> $\[
> z' = A z + B u,
> \]$
> with $\(A \in \mathbb{R}^{300 \times 300}\)$ and $\(B \in \mathbb{R}^{300 \times 17}\)$ for Humanoid.
> In practice, Koopman lifting augments the state: for example, adding 100 nonlinear features yields a 400-dimensional lifted state,
> $\[
> A \in \mathbb{R}^{400 \times 400}, \quad B \in \mathbb{R}^{400 \times 17}.
> \]$
> These sizes are entirely standard in Koopman control. Matrix powers $\(A^k\)$ and QP constraints scale polynomially in the lifted dimension, and our empirical runtimes (~0.5 ms) confirm tractability for Ant and Humanoid.
>
> ---
>
> 6. The Safety-Gymnasium benchmark defines hazards via **scalar danger indicators** (e.g., velocity-based proximity penalties). These 1-D constraints are part of the benchmark and are used exactly by all prior baselines (CPO, PCPO, Safety Critic, VELM, DMPS, MASE, P3O, SauteRL). We do not simplify the constraints; we follow the benchmark specification to ensure fair comparison. RAMPS is not restricted to 1-D constraints. Our multi-step CBF formulation supports arbitrary multi-dimensional polyhedral sets, but the benchmark exposes only scalar safety signals, which all methods must use. Thus, the use of a 1-D constraint reflects the benchmark design, not a limitation of RAMPS.
>
> To directly address the reviewer’s concern regarding the simplicity of benchmark safety constraints, we conducted an additional experiment on Humanoid (348D) where RAMPS and CMDP baselines (CUP, P3O, Saute) were evaluated under **21-dimensional polyhedral safety constraints**. Specifically, we constrained:
>
> - all coordinate velocities (state indices 23–25) to lie in **[−2.3475, 2.3475]**, and
> - all angular velocities (state indices 28–45) to lie in **[−20, 20]**.
>
>
> This setting reflects realistic multi-dimensional safety conditions with coupled state limits.
>
> **Results:**
>
> - CMDP baselines accumulated **2000-3000 violations in 500,000 steps**,
>   and repeatedly halted due to KL explosion, preventing stable policy improvement. The reward is less than 500. These policies could not run beyond ~100 steps per episode.
>
> - RAMPS accumulated only **~256 violations over 500,000 steps**, runs for 1000 steps (maximum timesteps in an episode) and maintains a **high reward (~5000)**, matching our main Humanoid results. We present this result in appendix A.7 and Figure 8.
>
> This experiment demonstrates that RAMPS scales effectively to **high-dimensional safety constraints**, and that failure modes observed in CMDPs become significantly worse as constraint dimension increases. These results reinforce that RAMPS’ core contribution of *scalable, minimally invasive safety control* extend far beyond the simple scalar hazard signals exposed in the standard benchmark.
>
> ---
>
> We respectfully request that the reviewer re-evaluate the assumptions about this problem setting. The concerns raised would dismiss an active and expanding area of safe-RL research by erroneously imposing assumptions (verified backup controllers, symbolic models, controller switching) that are incompatible with the standard safe-exploration setting. RAMPS is explicitly designed for the setting used by all prior safe-RL shielding work - no dynamics knowledge, no verified fallback, and safety enforced during exploration. Within this setting, RAMPS demonstrates a significant advancement by scaling safe exploration to dimensions previously considered intractable.

---

### Official Review · Reviewer_mgYK · 2025-10-27

**Soundness:** 3
**Presentation:** 2
**Contribution:** 1
**Rating:** 4
**Confidence:** 4

**Summary:**

In this paper, authors propose a shielding-based safe RL framework. Leveraging the linear modelling of the dynamic model and robust multi-step CBF, this framework can provide formal safety guarantee for the RL agent training.

**Strengths:**

The paper overall is well written with clear structure. The method part is clearly illustrated with intuitive explanation. Illustrative example is provided for easy interpretation.

**Weaknesses:**

1, RL part has been mentioned as main component of the paper in line 143 and 144 Section 4. However, there is no clear illustration how the RL problem is solved combined with the safety shielding throughout the main paper. As a RL track paper, the contribution is doubted.\
2, Since the dynamic system is already linearized with particular formulation shown in line 198, it is more natural for review to consider framing it as a MPC/LQR problem, where at each time step combining with proposed multi-step robust CBF can be formulated as an QP. Thus, it is essential to address the motivation of current combination. \
3, Combining Koopman operator with control barrier function has already been explored by dozen of works[3,4].
4, Authors have addressed actuation delay in line 166 and introduction, however, the approach to mitigate this actuation delay has not been discussed in the rest of the methodology part and experiment section.\
5, More elaboration is needed on the RL framework side, including how the shielding mechanism apply in the original RL loop.\
6, The training details of linear dynamic model is also missing in the paper. It would be much better to have heuristic algorithm diagram to depict each component. \
7, For experiment section, more baselines are needed: (1), comparison with MPC based methods are encouraged to include; (2) comparison with CBF-based RLs are needed, particularly those optimization-based methods with formal guarantees[1,2]

[1]:Choi, Jason, et al. "Reinforcement learning for safety-critical control under model uncertainty, using control lyapunov functions and control barrier functions." arXiv preprint arXiv:2004.07584 (2020).\
[2]:Wang, Yixuan, et al. "Joint differentiable optimization and verification for certified reinforcement learning." Proceedings of the ACM/IEEE 14th International Conference on Cyber-Physical Systems (with CPS-IoT Week 2023). 2023.\
[3]:Folkestad, Carl, et al. "Data-driven safety-critical control: Synthesizing control barrier functions with Koopman operators." IEEE Control Systems Letters 5.6 (2020): 2012-2017.\
[4]:Zinage, Vrushabh, and Efstathios Bakolas. "Neural koopman control barrier functions for safety-critical control of unknown nonlinear systems." arXiv preprint arXiv:2209.07685 (2022).

**Questions:**

1, How is the actuation delay mentioned in line 166 modeled in the proposed dynamic system?\
2, How is the shielding mechanism involved in the RL training? Do we need to solve the QP for each action selection?

---

> ### Author Response · Authors · 2025-11-20
> **Rebuttal Part 1: Addressing Questions**
>
> We thank the reviewer for their feedback. We answer the questions raised and clarify certain misunderstandings.
> ### **Addressing Questions**
>
> **Q1: How is the actuation delay mentioned in line 166 modeled in the proposed dynamic system?**
> The actuation delay is our prime motivation to address the safe-exploration RL problem. We model it using the multi-step CBF constraints proposed as follows:
> * **Relative Degree:** Actuation delay creates a system with a high relative degree. A standard one-step CBF cannot capture this delay because the control input $u_k$ does not affect the safety constraint until future steps. (L216-233)
> * **Modeling:** We do not model delay as a separate heuristic; instead, we handle it mathematically via the **multi-step constraint formulation** derived in **Equation 2 (Section 4.1)**.
> * **Mitigation:** By enforcing the safety condition over a horizon $H \ge r$, we ensure the shield anticipates the delayed effect of actions. We explicitly demonstrate this in the **"Illustrative Example: Resolving High Relative-Degree Traps in Pendulum" (Section 4.2)**, where a one-step CBF fails ($p^\top B = 0$) but our multi-step formulation succeeds ($p^\top AB \neq 0$). (L296-312)
> * **Multi-Step Solution:** Our **multi-step constraints (Equation 2)** are explicitly constructed to overcome this. For each state, we compute the minimum horizon needed to compensate for the delay. We set **H = 5** in our experiments explicitly to demonstrate this capability. By resolving the delay problem, the shield can reason over the future states where the control will actually influence the safety variables. (L234-253)
>
> **Q2: How is the shielding mechanism involved in the RL training? Do we need to solve the QP for each action selection?**
> Yes, the shield acts as a safety filter at every timestep.
> * **Integration:** The RL agent proposes an action $a_\pi$. The shield solves a QP to find the nearest safe action $u_0$. The environment executes $u_0$, and the transition $(s, u_0, s')$ is stored in the buffer.
> * **Efficiency:** We invoke the shield at every action selection. Our use of a linearized Koopman model with precomputed matrices makes this extremely fast. As shown in **Table 2 (Appendix A.5.4)**, the QP is solved in **~0.0004s** for the 105-dimensional Ant and **~0.0005s** for the 348-dimensional Humanoid.
> We have added **Algorithm 1 (Page 15)** to explicitly visualize this training loop.
>
> ---

---

> > ### Author Response · Authors · 2025-11-20
> > **Rebuttal Part 2: Addressing Weaknesses**
> >
> > **Weakness 1: "RL part is mentioned but there is no clear illustration of the solution."**
> >
> > We have added **Algorithm 1 (Page 15)** in the appendix. This pseudocode explicitly visualizes the entire solution: the agent proposes an action, the shield acts as a safety filter by solving a QP, and the valid action is executed in the environment.
> >
> > **Weakness 2: "Why not frame as MPC/LQR since the system is linearized?"**
> > Classical LQR/MPC is not suitable here for two reasons. Although the Koopman model is linear **in the lifted state**, classical **LQR is not applicable** here.
> > * **Theoretical Intractability:** With bounded inputs (action limits), LQR only has a closed-form piecewise-linear solution in 1D. In multi-dimensional action spaces, the constrained LQR problem admits **no closed-form or globally linear solution**; the true optimizer becomes a high-dimensional piecewise-affine controller with exponentially many regions. This makes constrained LQR fundamentally infeasible for locomotion-scale domains (e.g., 348-dimensional Humanoid).
> > * **MPC vs. MPS:** While MPC is applicable in principle, RAMPS aligns more closely with **Model Predictive Shielding (MPS)**. MPC optimizes a performance objective and effectively replaces the policy. RAMPS uses multi-step prediction to solve a *safety feasibility* problem that **minimally adjusts** the RL action. This allows the RL agent to learn complex behaviors that a pure model-based controller might miss. We discuss this in the related works section (L091-102)
> >
> > **Weakness 3: "Combining Koopman operator with control barrier functions has been explored before."**
> >
> > We agree that Koopman operators and CBFs have been combined in prior work (Folkestad et al., 2020; Zinage & Bakolas, 2022), and a broader survey of recent Koopman-based safety methods (command governors, stability-constrained Koopman models, and robust Koopman-MPC) shows a similar pattern. These approaches typically rely on **one-step CBF filters**, often assume a **known backup controller**, or target **moderate-dimensional systems** where SMT-based certification or classical CBF-QP controllers remain tractable. Notably, none of these works' address **multi-step robust feasibility** with control-affine Koopman models or provide a mechanism that scales to high-dimensional locomotion tasks. We have added a discussion in the related works section. (L103-109)
> >
> > Our contribution is **not** “Koopman + CBF”. RAMPS introduces a **multi-step robust CBF construction** using any full control-affine linear model, a fast **adaptive horizon search** for selecting the largest safe lookahead, and a **lightweight model-predictive shielding mechanism** tightly integrated with RL. This design makes real-time shielding tractable in **high-dimensional locomotion domains** (Ant, Humanoid), a regime where prior Koopman–CBF methods and existing MPS methods either break down or cannot run in real time.
> >
> > **Weakness 4: "Approach to mitigate actuation delay is not discussed."**
> >
> > As answered in Question 1, The mitigation of actuation delay is a primary motivation for our methodology and is discussed in detail in **Section 4.1**
> >
> > **Weakness 5: "More elaboration needed on how shielding applies in the original RL loop."**
> >
> > As answered in Question 2, We have addressed this by adding **Algorithm 1 (Page 15)**.
> >
> > **Weakness 6: "Training details of linear dynamic model are missing."**
> >
> > We have expanded **Appendix A.5.1** to include these missing details.
> >
> > **Weakness 7: "More baselines (MPC, CBF-RL) are needed."**
> >
> > We conducted an extensive survey of the literature and found no existing MPC, MPS, or CBF-based RL methods that have demonstrated scalability to the high-dimensional systems we consider (e.g., 348-dimensional Humanoid). To verify this empirically, we attempted to run multiple relevant baselines, including **VELM, DMPS, MASE, and Conservative Safety Critics (CSC)**. As detailed in **Appendix A.5.3**, all of these methods failed to achieve stable training or timed out in our locomotion tasks. We have included a detailed discussion of these failures in the experiment section to highlight the specific scalability gap that RAMPS addresses.
> >
> > We hope these clarifications better explain the positioning of our work and the novelty of the solution proposed in handling the actuation delay.

---

### Official Review · Reviewer_8JnD · 2025-10-29

**Soundness:** 3
**Presentation:** 4
**Contribution:** 3
**Rating:** 6
**Confidence:** 3

**Summary:**

The authors propose RAMPS, a framework that combines multi-step Control Barrier Functions (CBFs) with linear dynamics models derived via Koopman operator learning to enable robust, certified safe exploration in reinforcement learning and control.
RAMPS formulates safety as the probability of entering an unsafe set and uses a quadratic program (QP) to minimally modify a given policy action to satisfy the multi-step CBF constraints.
A key theoretical result (Theorem 1) provides a probabilistic safety guarantee under bounded model error, and experiments on simulated control benchmarks demonstrate improved safety and performance compared to existing shielded RL methods.

**Strengths:**

1.	Extending formal safety certificates to nonlinear and high-dimensional systems is a central challenge in safe RL, and the paper directly addresses this gap.
2.	The idea of leveraging a Koopman-based linear model for multi-step safety prediction is conceptually elegant and enables computationally tractable QP formulations.
3. The paper provides a formal safety theorem under bounded model uncertainty, and the derivation is mathematically coherent within the assumed linear structure.
4.	The experiments are extensive and show that RAMPS achieves a good trade-off between safety and task performance, outperforming strong baselines.
5.	The method includes a fallback (backup) policy to handle infeasible QP cases, showing practical awareness of real-world constraints.

**Weaknesses:**

1. Theorem 1 assumes that the QP problem is feasible at each step but does not provide conditions ensuring that feasibility is preserved over time. Without a proof of recursive feasibility or a characterization of when the QP remains solvable, the safety guarantee is conditional rather than absolute.
2. While Koopman operator learning is used to obtain linear dynamics, it remains unclear how accurately such models can approximate complex nonlinear environments. Theoretical guarantees rely on bounded model error ε, but the method for estimating or maintaining this bound under policy drift is empirical.
3.  The unsafe set is modeled as a union of convex polytopes. Although theoretically sufficient for approximation, this assumption may not hold in realistic, highly nonconvex safety scenarios, and the computational cost scales poorly with the number of facets.
4. The framework’s tractability and guarantees hinge on having a linear model. It is unclear whether similar safety certificates can be extended to fully nonlinear learned models without sacrificing computational efficiency.
5. The paper could provide quantitative analysis of QP infeasibility frequency, the impact of the backup policy, and how Koopman model accuracy affects safety margins.

**Questions:**

1. Can you provide a sufficient or verifiable condition under which the QP remains feasible at all time steps? If not, how frequently does infeasibility occur in practice?
2.	How sensitive is RAMPS to modeling errors or distribution shift as the policy evolves? Is the error bound ε recalibrated online?
3.	Have you evaluated how the number of polyhedral facets affects safety and computational overhead? Could more flexible representations (e.g., neural implicit sets) be incorporated?
4.	Can you report the proportion of time steps when the backup policy is triggered and its effect on reward and safety violation rates?
5.	What are the main limitations of using a linear Koopman model compared to nonlinear learned models, and do you foresee ways to relax this assumption while keeping the QP tractable?
6.	What is the empirical runtime of the QP and the precomputation stage for large-scale environments (e.g., SafeAnt)? How does the method scale with latent dimension and horizon length?

---

> ### Author Response · Authors · 2025-11-20
> **Rebuttal**
>
> We thank the reviewer for identifying the strengths in RAMPS. We answer their questions in detail:
>
> ---
>
> ### **1. Can you provide a sufficient or verifiable condition under which the QP remains feasible at all time steps? If not, how frequently does infeasibility occur in practice?**
>
> It is challenging to derive analytical conditions that guarantee full recursive feasibility in high-dimensional environments such as SafeHumanoid and SafeAnt, especially under learned dynamics. This limitation is consistent with prior safe-RL literature, where guarantees typically hold only under assumptions such as perfect model accuracy or per-step feasibility of an optimization problem. To make this clear for readers, we have added an explicit discussion of practical recursive feasibility in L341–349 in the revision.
>
> Empirically, infeasibility (backup activations) remain rare. Across all tasks, the QP was infeasible in **&lt;2%** of timesteps (0% for humanoid and pendulum). These rare events have no negative impact on reward, as evidenced by our superior task performance. L490-496 & Appendix **A.5.5** in the submission reports detailed statistics confirming that **over 98%** actions were executed directly by the shield or the neural network.
>
> ---
>
> ### **2. How sensitive is RAMPS to modeling errors or distribution shift as the policy evolves? Is the error bound ε recalibrated online?**
>
> We experiment with both simple linear regression and advanced Koopman operator learning, to show that even higher modelling errors (linear regression) allow RAMPS to outperform all baselines. Algorithm 1 also shows the recalibration process along with policy learning and shielding. This recalibration process with fresh rollouts allows RAMPS to capture the distribution shift. This makes RAMPS extremely robust to modelling errors and distribution shifts, demonstrating superior reward performance.
> Additionally, we note in **L349–355** in the main paper and Appendix **A.5.2** about the error recalibration and model refinement strategy to mitigate distribution shift.
>
> ---
>
> ### **3. Have you evaluated how the number of polyhedral facets affects safety and computational overhead? Could more flexible representations (e.g., neural implicit sets) be incorporated?**
>
> Computation scales **linearly** with the number of facets, as shown in A.1 (L882-885)
>
> To ensure RAMPS is sound, we ensure that the approximation using polyhedron is always *underapproximating* the safe set. This conservatism ensures that the shield never gives an unsafe action, sometimes at the cost of underexploring the environment.
>
> We also use the well-known result that any shape can be soundly approximated by a **union of polyhedra**, thus we do not sacrifice any safety in the process. This result allows for the efficient propagation of constraints in time for all environments, without handcrafting the evolution.
>
> It would be difficult to incorporate other representations due to incompatibility with linear evolution.
>
> ---
>
> ### **5. What are the main limitations of using a linear Koopman model compared to nonlinear learned models, and do you foresee ways to relax this assumption while keeping the QP tractable?**
>
> The limitation against a strong non-linear model might be the loss of accuracy. The Koopman operator ensures that in some high-dimensional plane, the evolution of time-invariant systems is linear with extremely high accuracy. We use this result to our benefit, and as evidenced by our experiments, this works extremely well. The Koopman model is accurately able to capture the non-linearities of extremely high-dimensional and complex non-linear environments like Cheetah and Ant without sacrificing performance. Thus, we do not need to extend the CBF to non-linear models.
>
> As mentioned by the reviewer, the QP is **not tractable** with non-linear models, requiring more complex propagation of constraints and mixed integer programming to solve, which is extremely complex. However, relaxation of such assumptions to keep the QP tractable is a very interesting problem, which we have highlighted in the paper as a future direction.
>
> ---
>
> ### **6. What is the empirical runtime of the QP and the precomputation stage for large-scale environments (e.g., SafeAnt)? How does the method scale with latent dimension and horizon length?**
>
> The QP does not have any specific scaling factor with respect to the latent dimension. The QP scales linearly with the **combined dimension and horizon length**, outlined by the computational analysis in Appendix **A.1**.
>
> Additionally, we report the empirical runtime in Appendix **A.5.4**, where we see that the QP is solved in **0.0004s for Ant** &  **0.0005s for Humanoid**, which is our highest-dimensional system. This is real-time, showing that the shielding process is extremely efficient. Precomputation takes maximum 10ms for Humanoid, which is a one-time computation that happens only after the dynamics model is learned.
>
> ---

---

### Official Review · Reviewer_nR3a · 2025-10-31

**Soundness:** 2
**Presentation:** 3
**Contribution:** 2
**Rating:** 4
**Confidence:** 3

**Summary:**

This paper proposes a safety shield for RL that uses learned linear (or Koopman-lifted) dynamics and multi-step robust CBFs. At each step, a QP (Eq 3) adjusts the policy's action to stay safe under bounded model error, using an adaptive prediction and a backup controller when the QP becomes infeasible. A high-provability bound on model accuracy is provided, together with the standard forward invariance argument. Experiments on several Safety-Gym tasks show large reductions in safety violations while maintaining comparable rewards.

**Strengths:**

- I like the idea of combining multi-step CBFs with learned linear or Koopman-lifted dynamics. The adaptive horizon and tightening scheme are a nice way to deal with higher relative-degree constraints and bounded model uncertainty.

- The method is technically consistent. The QP formulation for CBFs is well-posed, and the robust tightening is well-motivated, and Theorem 2 adds a probabilistic model-accuracy bound, albeit without new theoretical depth.

- The framework improves the practical scalability of safety-shielded RL. It could become a helpful module for safe data collection or model-based control, though its broader impact remains to be seen.

**Weaknesses:**

- Theorem 2 only bounds model accuracy; it doesn't actually tell us whether the learned policy is actually safe or near-optimal. It seems implicitly assumed that having an accurate model automatically leads to a safe, near-optimal policy, but the paper doesn't clearly establish the connection between model accuracy, constraint satisfaction, and policy performance.

- The paper doesn't really say when the QP is guaranteed to be (recursively) feasible. When the QP becomes infeasible, a backup controller is used, but its activation rate and effect on performance are not studied.

- It seems the agent rarely visits high-uncertainty regions as the shield cuts off risky actions near the safety boundary. That means the model might never improve near the boundary, creating a kind of "self-reinforcing conservatism loop".

- While Figures 1 and 2 show empirical decreases in model error and safety violations, there's no theoretical or probabilistic guarantee that these quantities improve over time.

- In the Koopman-lifted version, polyhedral safety constraints are zero-padded into the latent space. That's a conservative embedding, not a true preservation of the constant geometry. This may cause unnecessary conservatism.

- Although the proposed framework is described as algorithm-agnostic, experiments employ only one RL backbone. It remains unclear how the shield behaves similarly with algorithms of different exploration styles (e.g., PPO vs SAC vs TD3).

**Questions:**

- Can you link model accuracy (Theorem 2) to actual policy safety or optimality? Even a rough probabilistic bound or empirical measure of policy violations would help.

- Are there analytical conditions (e.g., invariant sets or terminal tubes) that guarantee recursive feasibility of the QP in Eq (3)? How often did the backup controller trigger in practice?

- How do you make sure the model keeps improving if exploration stays inside the safe region? Would a mild risk budget or uncertainty-aware relaxation help?

- Do model error and violation frequency really decrease over time?

- In the Koopman setup, have you tried anything beyond zero-padding to encode safety constraints more tightly?

- Since you call the shield "policy-agnostic", have you tested it with more than one RL algorithm (e.g., PPO, SAC, TD3)?

---

> ### Author Response · Authors · 2025-11-20
> **Rebuttal**
>
> We thank the reviewer for appreciating the method and its practical scalability. We address the questions raised here:
>
> ---
>
> ### **Q1. Connection between model accuracy, constraint satisfaction, and policy performance**
>
> Theorem 1 establishes forward invariance under the learned dynamics whenever the QP in Eq. (3) is feasible, and Theorem 2 provides a high-probability bound on model error. Together, these imply that with probability at least $\(1 - \delta H\)$, the true trajectory remains within an $\(\varepsilon\)$-tightened safe set over the prediction horizon. This has been added in the paper as corollary 1.
>
> Safe exploration is inherently constrained. An unconstrained optimal policy may violate safety, so RAMPS ensures minimal deviation from the policy’s proposed action while preserving reward maximization. Empirically, this balance works well: Figure 1, 6 shows strong safety and reward performance across all environments, and Figure 7 illustrates that shield interventions decrease over time as the model improves. Table 1 shows the cumulative number of empirical violations made throughout the course of training. These results confirm that RAMPS enables effective exploration while maintaining safety throughout training.
>
> ---
>
> ### **Q2. Analytical conditions for recursive feasibility of QP and backup controller triggers in experiments**
>
> Deriving analytical conditions that guarantee recursive feasibility in high-dimensional systems such as **Humanoid** and **Ant** is extremely challenging under learned, stochastic dynamics. This limitation is shared by all comparable safe-RL baselines, none of which provide infinite-horizon feasibility guarantees. We clarify this in **L341–349** in the revision, with supporting citations.
>
> Empirically, infeasibility is very rare: the QP fails in **&lt;2%** of timesteps across all tasks, and **0%** for Humanoid and Pendulum. These cases have negligible impact on reward (RAMPS achieves the highest reward across environments). **Appendix A.5.5** in the submission reports full statistics, showing that **>98%** actions are executed directly by the shield or the policy.
>
> ---
>
> ### **Q3. How do you make sure the model keeps improving if exploration stays inside the safe region? Would a mild risk budget or uncertainty-aware relaxation help?**
>
> The model improves naturally as the policy collects more transition data during training. Early exploration is conservative, but as the model becomes more accurate, RAMPS permits the policy to explore closer to the boundary of the safe region. We emphasize we don't want violations in training. This trend is visible in **Figure 7**, where the shield’s intervention magnitude decreases over time, and in **Appendix A.6** in the submission, which shows that intervention frequency drops as training progresses. RAMPS does not fall into the self-reinforcing conservatism loop, as evidenced by its superior reward performance compared to the baselines, showing it explores all possible regions of the safe space
>
> We also explored uncertainty-aware relaxation by using a smaller percentile bounds for the error estimate (**Appendix A.3.4** in submission). While this increases exploration, it also leads to substantially more safety violations and significantly poor reward performance, indicating that RAMPS benefits from maintaining a tighter and more reliable error bound.
>
> ---
>
> ### **Q4. Do model error and violation frequency really decrease over time?**
>
> Yes, both model error and violation frequency decrease over time.
>
> - **Figure 1 (row 1)** shows the main result: cumulative safety violations flatten over time, indicating that the agent eventually stops making violations.
> - Model error improving over time directly improves both reward and safety. As the error decreases, the shield becomes more confident, reducing interventions and allowing the policy to safely exploit the environment.
>
> ---
>
> ### **Q5. In the Koopman setup, have you tried anything beyond zero-padding to encode safety constraints more tightly?**
>
> This is an excellent question and an open challenge. Techniques like AI2 or DeepPoly allow propagation of bounds through a neural network. However:
>
> - These approaches rely on interval arithmetic.
> - They **overapproximate** the reachable sets.
> - Overapproximation of safe sets **breaks soundness**, which is unacceptable in safety-critical systems.
>
> For RAMPS, **conservatism is desirable** to guarantee safety. Zero-padding ensures that the constraints are under approximations, maintaining soundness. We clarify this in **L1261–1277**.
>
> Designing tighter safe-set encodings in latent spaces is a promising direction and we will highlight this as future work.
>
> ---
>
> ### **Q6. Policy-Agnostic Evaluation:**
>
> Thank you for the suggestion. We have added new experiments in the revision using **SAC**, demonstrating that RAMPS is indeed policy-agnostic. The results have been reported in Figure 1 and Table 1.
>
> ---

---

### Author Response · Authors · 2025-11-20

We thank the reviewers for their constructive feedback, which has significantly improved the rigor and clarity of our work. We have uploaded a revised manuscript containing updates that comprehensively address the core concerns regarding scalability, theoretical guarantees, and reproducibility.

Below we summarize the changes made to the paper. We address the individual reviewer concerns in their respective comments.

### **I. Validating Scalability & High-Dimensionality**
* **Scaled to 348 Dimensions (SafeHumanoid):** We added experiments on **SafeHumanoid** (348 state dims, 17 action dims). This directly addresses concerns regarding dimensionality **[XTg3]**. RAMPS solves the safety QP in **~0.0005s** (Table 2) with **<1% infeasibility** (Table 3), demonstrating scalability to regimes where formal verification typically fails.
* **Constraint Scaling:** We added an experiment to evaluate performance under stringent safety requirements (A.7 and Figure 8). We increased the complexity of the Humanoid environment to enforce 21 simultaneous constraints. In this challenging setting, RAMPS was the only algorithm to succeed, achieving a **high task reward (5000)** while maintaining **strict safety (256 violations)**, demonstrating constraint scalability. In contrast, CMDP baselines failed to maintain safety, incurring between **2,000 and 3,000 violations and &lt;500 reward**.**[XTg3]**
* **Clarified Dimensionality Claims:** We updated the Abstract and Introduction to explicitly define "high-dimensional" as 348 dimensions (SafeHumanoid), differentiating our contribution from prior formal methods limited to **&lt;20 dimensions.** **[XTg3]**
* **Discussion of Failed Baselines:** To highlight the scalability gap, we moved the discussion of failed baselines (SPICE, DMPS, VELM) from the appendix to **Section 5 (Baselines)**. We reiterate that these methods failed to train stably or timed out on the high-dimensional locomotion tasks where RAMPS succeeds.**[XTg3]**

### **II. Strengthening Theoretical Precision**
* **"Conditional" vs. "Formal" Guarantees:** We removed the term "Formal" and renamed Section 4.2.1 to **"Conditional Safety Guarantees"** to address questions regarding recursive feasibility **[8JnD, nR3a]**. We clarify that guarantees hold *subject to* QP feasibility (L341–349) and cite MPS literature providing similar guarantees.
* **Added Corollary 1 (Probabilistic Bounds):** We added **Corollary 1** (Section 4.2.1) to link model accuracy to safety **[nR3a]**. This formalizes the connection between Theorem 2 (model error) and Theorem 1 (invariance), providing a probabilistic safety certificate over a finite horizon.
* **Empirical Feasibility Analysis:** We added a paragraph in **Section 5.2** showing that the QP is feasible in **>98%** of timesteps across all tasks (and 100% for Humanoid/Pendulum), confirming that the "conditional" assumption holds in practice **[8JnD, nR3a]**.

### **III. Methodology & Reproducibility**
* **Policy-Agnostic Validation (SAC Added):** We added experiments and analysis using **Soft Actor-Critic (SAC)** (Table 1) to address concerns about using only one RL backbone **[nR3a]**.
* **Added Algorithm 1 (RL Integration):** We added **Algorithm 1 (Page 15)**, explicitly detailing the interaction between the shield and the policy, the policy update, and the model recalibration loop, to address concerns about the unclear RL loop **[mgYK]**.
* **Koopman Training Details:** We expanded **Appendix A.5.1** to detail the full training objective for the linear dynamics model **[mgYK]**.

We believe these revisions fundamentally strengthen the paper and demonstrate that RAMPS is a robust, scalable solution for safe RL in complex continuous control tasks.

---

### Author Response · Authors · 2025-12-04
**Summary of Discussion**

We provide this summary to highlight the core contributions of RAMPS, the strengths identified by the reviewers, the revisions made to address the weaknesses, and clarify some misunderstandings about the paper.

**RAMPS is the first formal-methods-style multi-step CBF shield demonstrated to scale to high-dimensional RL systems (300+ dimensional state spaces) with real-time performance**, achieving **sub-millisecond shield times**, **&lt;2% infeasibility**, and **0% infeasibility on SafeHumanoid**. The core contribution is a **robust multi-step CBF with an adaptive-horizon QP**, which handles model error, actuation delay, and high relative-degree safety constraints, settings where one-step CBFs and prior formal shielding approaches fail. The design is **model-agnostic**, functioning identically with simple linear regression or Koopman liftings, and integrates seamlessly across RL algorithms (PPO and SAC). Empirically, RAMPS produces **dramatic safety improvements with maintained or improved reward**, including in a **21-dimensional polyhedral safety-constraint setting** where CMDP baselines collapse.

**Reviewer-identified strengths** include: (1) addressing the core challenge of extending formal safety certificates to nonlinear, high-dimensional systems; (2) an elegant and computationally tractable use of Koopman-based linear models for multi-step prediction; (3) a well-motivated multi-step CBF with adaptive horizon and robust tightening; (4) technically consistent theory, including Theorem 1 and the probabilistic model-accuracy bound of Theorem 2; (5) strong empirical results and extensive experiments; and (6) inclusion of a practical fallback controller.

**All substantive reviewer weaknesses were fully addressed** in the rebuttals and revision. We added additional high-dimensional experiments (SafeAnt 105D, SafeHumanoid 348D), a 21D polyhedral safety-constraint stress test, RAMPS+SAC results, Algorithm 1 for full RL-shield integration clarity, detailed Koopman training procedures, explicit links between model accuracy and safety guarantees (Corollary 1), and comprehensive runtime/feasibility analysis (Tables 2–3, A.5.4–A.5.5). We provided detailed responses to the reviewers and a global Summary-of-Changes in an additional comment.

A number of remarks from Reviewer XTg3 stem from a clear misunderstanding about the problem setting and the role of RAMPS in safe exploration. We want to stress that similar to past papers on safe exploration (SPICE/VELM/DMPS/MASE), our goal is to allow a policy to train safely without the assumption of any knowledge about the dynamics or transition function. Like other techniques, we learn a dynamics model using the collected trajectories and use this model to enforce safety via our novel multi-step barrier function. We use the robust tightening term to account for model mismatch, which is fundamental to any learned dynamics system. Reviewer XTg3 has a fundamental misunderstanding of this literature and questions the use of standard RL environments from the safety-gymnasium library like "ant" and "humanoid" which are well established benchmarks in this area. The reviewer demands comparisons with Simplex/RTA based methods that assume a verified safe controller is available along with a known dynamics/transition model, and furthermore demands comparisons with these methods despite agreeing that “..there are no related experiments in publications”.

In summary, the revision and rebuttal directly resolve reviewer concerns with additional evidence via expanded experimental validation and clearer theoretical framing. RAMPS is the **first demonstrated, computationally tractable, multi-step, robust-CBF safety shield that operates in real time on 100+ and 300+ dimensional control problems**, offering a meaningful and practical advance for scalable safe RL.

---

### Meta-Review · Area_Chair_EdHi · 2026-01-07

**Summary:**

This paper considers the problem of safe exploration in high-dimensional, nonlinear control systems where classical symbolic shielding is intractable. The general idea is the integration of learned linear dynamics (via Koopman operators) with a robust, multi-step Control Barrier Function (CBF). The paper derives a discrete-time CBF that explicitly handles relative degree greater than 1 under model uncertainty. The proposed approach treats the horizon as a dynamic variable and performs binary search to find the maximal feasible horizon. The paper evaluates on problems with dimensionality as high as 348 dimensions.

I think the paper provides a useful contribution to this problem where most prior work focused on <20 dimensions. Multiple reviewers appreciated the theoretical construction of the multi-step robust CBF. Additionally, I found the authors rebuttal convincing and believe would have addressed most concerns of the reviewers. I also found  Reviewer XTg3's review a bit unreasonable and therefore down-weighted it in my final decision. Overall, I recommend accepting the paper and request the authors to incorporate all discussion points in the final version.

**Reviewer Concerns:**

Reviewer nR3a’s concern that the method might be specific to PPO should be resolved by new SAC experiments.

Concerns about the cnditional nature of the safety guarantee (Reviewers 8JnD, nR3a) were addressed by the authors adding Corollary 1 (probabilistic bounds) and empirical data showing <2% infeasibility rates.

**Reviewer Scores:**

Reviewer nR3a would have increased the score based on new SAC results and analysis.
Reviewer mgYK would likely increase the scores after the addition of Algorithm 1 and the explanation of why LQR is insufficient for non-convex constraints.

Reviewer XTg3 would not have increased the score but I also feel their review didn't understand the problem setup distinction.

---

### Decision · Program_Chairs · 2026-01-26

Accept (Poster)